# ReBaR: Reference-Based Reasoning for Robust Human Pose and Shape Estimation from Monocular Images

## Abstract

This paper introduces a novel method, ReBaR (**Re**ference-**Ba**sed **R**easoning for Robust Human Pose and Shape Estimation), designed to estimate human body shape and pose from single-view images. ReBaR effectively addresses the challenges of occlusions and depth ambiguity by learning reference features for part regression reasoning. Our approach starts by extracting features from both body and part regions using an attention-guided mechanism. Subsequently, these features are used to encode additional part-body dependencies for individual part regression, with part features serving as queries and the body feature as a reference. This reference-based reasoning allows our network to infer the spatial relationships of occluded parts with the body, utilizing visible parts and body reference information. ReBaR outperforms existing state-of-the-art methods on two benchmark datasets, demonstrating significant improvement in handling depth ambiguity and occlusion. These results strongly support the effectiveness of our reference-based framework for estimating human body shape and pose from single-view images.

## 1 Introduction

Human body shape and pose estimation from monocular images has become increasingly important in computer vision due to its wide range of applications in fields such as human-computer interaction, virtual reality, and digital human animation. This task involves reconstructing the human body by obtaining parameters of a human body model (such as SMPL (Loper et al., 2015) and GHUM (Xu et al., 2020)) from single RGB images. Regression-based methods (Guler & Kokkinos, 2019; Kanazawa et al., 2018; Omran et al., 2018; Pavlakos et al., 2018) that predict human model parameters from image features have made significant progress in recent years and have become the leading paradigm. However, they still face several limitations.

One major limitation is the handling of severe occlusions. Occlusion occurs when a body part is obscured by another object or body part, making it difficult to estimate its pose accurately. Most existing methods based on global feature learning usually struggle with occlusions, which limits their performance in real-world scenarios. Recent work (Kocabas et al., 2021a) has attempted to alleviate this issue by leveraging neighboring visible parts to improve the estimation of occluded parts. However, this strategy is unreliable as it may mistake the pose prediction of an adjacent visible part as the occluded one in some cases, for example, when a leg is impeded mainly by one arm. Another limitation is the handling of depth ambiguity. Depth ambiguity occurs when the relative depth of two body parts cannot be determined from a single image, leading to errors in pose estimation. This is particularly challenging in monocular RGB settings together with self-occlusion. For example, as shown in Figure 1, given a human with hands behind the back, existing methods fail to estimate a reasonable pose of the forearms.

However, humans can effortlessly deduce that the hands are behind the back in such an image by utilizing a combination of whole-body visual cues. This suggests that humans rely on a form of reference-based reasoning when perceiving objects in images to handle conclusion and depth ambiguity. Inspired by human ability, we introduce the ReBaR framework. It uses reference-based reasoning to combine global visual cues for better pose estimation. This approach effectively han-

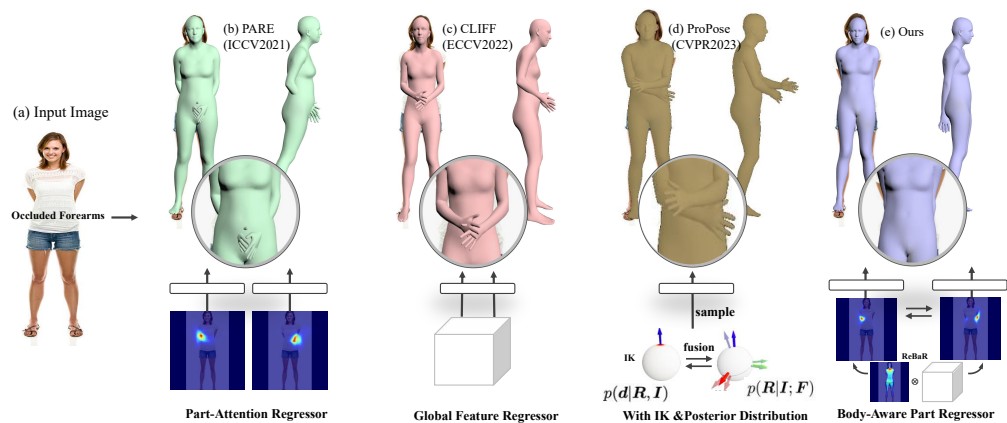

Figure 1: **Illustration of our method's main superiority over existing approaches.** When given a challenging input (a), PARE (b) (Kocabas et al., 2021a), CLIFF (c) (Li et al., 2022) and ProPose (d) (Fang et al., 2023) struggle to accurately estimate the pose of the forearm due to occlusion and depth ambiguity. Our method (e), on the other hand, tackles these challenges by exploring reference - based reasoning for part regression.

dles occlusions and depth ambiguity, leading to accurate human body shape and pose estimation from single-view images, even in difficult situations.

Our method primarily involves an attention-guided encoder for feature extraction and a body-aware regression module for part representation. The attention-guided encoder, based on ConvNets, focuses on body and part features while considering 2D and 3D dependencies between parts. The body-aware regression module uses a two-layer transformer to encode body-aware part features for each part and a one-layer transformer for per-part regression. This approach allows our method to incorporate both local visual cues and global information from the entire body, improving estimation accuracy by inferring the spatial relationship of occluded parts with the body.

Our experiments first show that our method consistently outperforms state-of-the-art approaches on 3DPW (Von Marcard et al., 2018) and Human3.6M (Ionescu et al., 2013) datasets. Then, we evaluate our method on 3DPW-OCC (Von Marcard et al., 2018) and 3DOH (Zhang et al., 2020b) datasets and demonstrate its effectiveness in occlusion handling. We also compare axis MJE on the 3DPW-TEST dataset. The result shows that our method's performance gain over state-of-the-art methods is attributed mainly to the much lower error of Z-axis MJE, demonstrating that it effectively reduces depth ambiguity in estimation. Finally, our qualitative results and ablation study confirm the proposed method's effectiveness.

## 2 RELATED WORK

Our proposed method is a regression-based framework for 3D human body reconstruction from a single RGB image, which can effectively tackle the issues of occlusion and depth ambiguity. Therefore we review the related works that can be divided into regression-based methods, occlusion handling, and reducing depth ambiguity.

### 2.1 REGRESSION-BASED METHODS

Estimating human mesh and pose from single images has attracted increasing attention in the computer vision community. The mainstream methods in this task include optimization and regression-based. Optimization-based approaches concentrate on the optimization algorithms to fit parametric models (e.g., SMPL (Loper et al., 2015)) based on 2D observations such as keypoints and silhouettes. These methods have been criticized for having long optimization processes and high sensitivity to initialization (Kolotouros et al., 2019). Therefore, regression-based approaches have become a more popular paradigm in recent years, as they can directly and fast regress parameters from im-

age features. HMR (Kanazawa et al., 2018) is a milestone work that introduces reprojection loss of keypoints as weak supervision, enabling training regression models on in-the-wild datasets. Recent work continues to explore more advanced weak supervision loss, such as introducing fine-grained correspondences (Omran et al., 2018; Rueegg et al., 2020; Xu et al., 2019; Zhang et al., 2020a), providing pseudo 3D ground truth (Kolotouros et al., 2019; Joo et al., 2020), and improving reprojection loss (Kocabas et al., 2021b; Li et al., 2022). For instance, SPIN (Kolotouros et al., 2019) incorporates the optimization process into the regression framework, allowing better-optimized 3D results as supervision for the network. SPEC (Kocabas et al., 2021b) estimates the perspective camera from a single image and utilizes it to improve reprojection loss. CLIFF (Li et al., 2022) proposes considering the box information and computing the reprojection loss from the original images, which facilitates predicting global rotations. However, all these methods employ global representation for regression, making them sensitive to occlusion and partially visible humans.

## 2.2 Occlusion handling

Handling occlusion is challenging yet crucial for human shape and pose estimation. Simulating data containing occlusion is a straightforward way towards this issue. This line of work attempts to generate training data of occluded human bodies by cropping images (Biggs et al., 2020; Joo et al., 2020; Rockwell & Fouhey, 2020), overlaying objects (Georgakis et al., 2020; Sárándi et al., 2018), and augmenting feature maps (Cheng et al., 2020). Despite the success achieved in reducing occlusion sensitivity, the realisticness of synthetic data is usually poor, resulting in unsatisfactory performance in dealing with real occlusion data. The other line of work relies on visibility cues to aid in dealing with occlusion. Cheng et al. (2019) obtains the keypoint visibility label and masks the loss of invisible points in training, encouraging the network to infer occluded joints from visible parts. Yao et al. (2022) proposes VisDB, which first trains a network to predict the coordinates and visibility label of the mesh vertex and then incorporates them to regress the SMPL parameters. However, VisDB is a two-stage framework and requires a test-time optimization procedure to obtain the final results. In contrast, PARE (Kocabas et al., 2021a) is an end-to-end learning framework that infers the occluded parts by exploiting the attention mechanism to find helpful information from neighboring visible parts. However, the visual cues of adjacent visible parts sometimes are unreliable or insufficient to infer the occluded parts. Unlike these approaches, Our method leverages visible parts and body reference information to infer the occluded parts.

## 2.3 Reducing depth ambiguity

Depth ambiguity is inherent in estimating 3D pose from a single RGB image. To address this issue, Wang et al. (2019) predicts a set of pose attributes that indicate the relative location of a limb joint with respect to the torso, which provides an explicit prior to reduce depth ambiguity. Both HybrIk (Li et al., 2021) and ProPose (Fang et al., 2023) methods leverage the predicted 3D keypoints. In particular, ProPose extends this foundation by incorporating skeletal orientation, thereby indirectly introducing depth information into the model. As a result, these approaches demonstrate high sensitivity to the accuracy of keypoint detection. Yao et al. (2022) predicts the vertex's visibility in the depth axis to resolve depth ordering ambiguities. The capability of these methods to deal with depth ambiguity mainly relies on the prediction of depth-related attribute labels. However, these label estimators are reasoned with local visual cues, suffering significant instability. For example, perturbation of local pixels will lead to wrong label predictions. In contrast, our method can effectively reduce the ambiguity of parts in depth space by learning a prior of part poses referring to the body.

## 3 Approach

**Preliminaries.** Our approach utilizes the SMPL parametric model to represent the human body. The model requires two essential characteristics: pose, denoted as $\theta \in \mathbb{R}^{72}$, and shape, denoted as $\beta \in \mathbb{R}^{10}$. The SMPL model produces a positioned 3D mesh $\mathcal{M}(\theta, \beta) \in \mathbb{R}^{6890 \times 3}$ as a differentiable function. The reconstructed 3D joints are obtained as $\mathcal{J}_{3D} = \mathcal{W}\mathcal{M} \in \mathbb{R}^{J \times 3}$, where $J = 24$ and $\mathcal{W}$ is the pre-trained linear regression matrix.

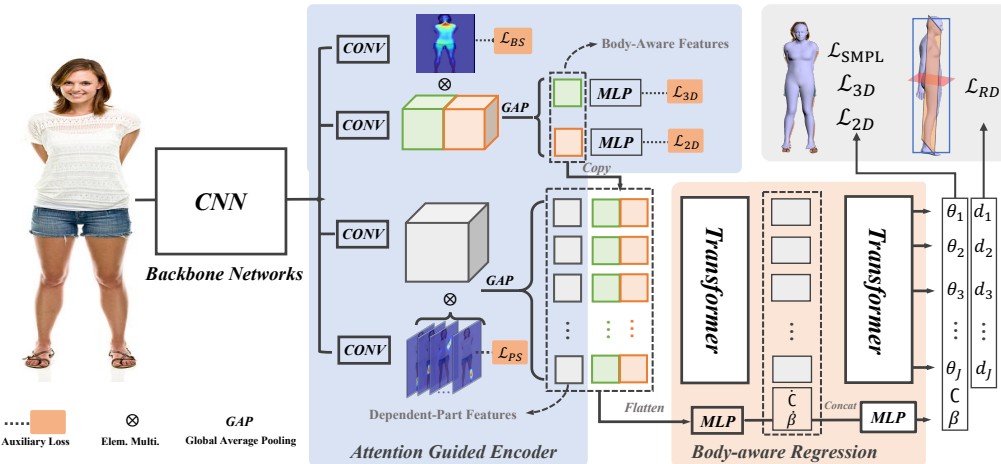

Figure 2: **ReBaR network architecture.** Given an input image, our method first extracts body and part features based on a soft attention mechanism. Then each part feature is concatenated with the body feature as an input token to the transformer to encode body-aware part features for camera prediction and SMPL parameter regression.

As depicted in Figure 2, our method mainly includes an attention-guided encoder and a body-aware regression module. The first component aims to learn proper attention maps to locate informative regions for extracting parts and body features. The second module encodes body-aware part features based on these output features and feeds them into a one-layer transformer for camera and SMPL parameter regression. We introduce our method in more detail as follows.

### 3.1 ATTENTION-GUIDED ENCODER

The human body is a complex structure with various parts and joints that have different contributions to the overall shape and pose. To accurately model and reconstruct the 3D human body from a single RGB image, it is crucial to focus on the informative regions and capture both global and local features. The attention-guided encoder (AGE) is designed to achieve this by learning proper attention maps and extracting features from different body parts and the whole body. We use a CNN backbone to extract feature volumes from the input image. From there, we leverage four convolutional layers to obtain the body attention map, body feature volume, part attention map, and part feature volume, respectively.

**Body-Aware Features.** The first task of the AGE module is to extract Attention-guided reference features. To achieve this, we pass the features extracted by the backbone network through a convolutional layer to obtain the body attention map $Att_{\text{body}} \in \mathbb{R}^{H \times W \times 1}$, which is supervised by the body segmentation map. Next, we use two additional convolutional layers to obtain a 2D feature $F_{2D} \in \mathbb{R}^{H \times W \times C}$ and a 3D feature $F_{3D} \in \mathbb{R}^{H \times W \times C'}$ (where $C$ and $C'$ are feature the dimensions of the 2D feature block and the 3D feature block respectively). These volumes help encode more global features and establish dependencies between 2D and 3D. To aid learning, we use global 2D/3D keypoint information for supervision. Then, we concatenate the two learned features to construct a global reference feature block $F_{\text{body}} \in \mathbb{R}^{H \times W \times (C+C')}$. Finally, we extract Attention-guided reference features $F_{\text{brf}} \in \mathbb{R}^{(C+C') \times 1}$ by performing Hadamard operations on the body attention map and the global reference feature block:

$$F_{\text{brf}} = \sigma(Att_{\text{body}})^T \odot F_{\text{body}} \tag{1}$$

where $\odot$ is the Hadamard product and $\sigma(Att_{\text{body}})$ is used as a soft attention mask to aggregate features. By using this approach, we can better capture the global features of the human body and establish the relationship between 2D and 3D.

**Dependant-part Features.** Another task of the AGE module is to extract part-query features. To accomplish this, we feed the feature volume extracted by the backbone network to two convolutional layers, which respectively extract part-3D features $F_{\text{part}} \in \mathbb{R}^{H \times W \times C''}$ and part-attention maps

$Att_{\text{part}} \in \mathbb{R}^{H \times W \times 24}$ (where $C''$ is the feature dimensions of the part-3D feature block). Follow the Pare method Kocabas et al. (2021a), we use part-segmentation maps as an auxiliary supervision to aid the learning process. Similar to the extraction process for Attention-guided reference feature, we perform a Hadamard operation on the part-feature block and the part-attention map to extract part-query features $F_{\text{pqf}} \in \mathbb{R}^{C'' \times 24}$. This approach allows us to focus on the relevant parts of the body and extract features that are important for accurate 3D reconstruction.

## 3.2 BODY-AWARE REGRESSION

The relationships between different body parts play a vital role in understanding the human body's 3D structure and pose. A simple concatenation of part features may not be sufficient to capture these relationships and dependencies. The body-aware regression module (BAR) is introduced to encode body-aware part features for each part while establishing associations between parts. This module helps the model to leverage the spatial relationships between body parts and the whole body, leading to a more accurate 3D reconstruction. These features are then utilized as input to the regression transformer, which predicts SMPL model parameters, camera parameters, and relative torso plane depth. Finally, the model parameters in the output are fed into the SMPL model to generate a 3D human body.

To encodes body-aware part features for each part, we concatenate the Attention-guided reference feature from the AGE module to the part-query features and establish a set of tokens $T = \{t_{bap}^1, t_{bap}^2, t_{bap}^3, ..., t_{bap}^{23}, t_{bap}^{24}\}$(where $t_{bap}^i = F_{pqf}^i \oplus F_{brf}$, $\oplus$ means concatenate operation.) as the input of the two-layer body-aware transformer. Using this encoder, we query part features from body feature to encode body-aware part features $F_{\text{bapf}} \in \mathbb{R}^{24 \times 128}$, building part representations and dependencies to reference body. The resulting body-aware part features sequence can not only capture the unique characteristics of each part but also combine relevant global information from the entire body, which is crucial for accurate 3D reconstruction. By using the transformer to establish dependencies between the parts and reference body, our feature encoder can effectively exploit the spatial relationship between parts and bodies to generate more informative part representations. Overall, our feature encoder is capable of finding useful information from parts and the whole body and outputs a set of body-aware part features that are highly informative for the subsequent regression task. Specifically, the formula is as follows:

$$Q = W^q T; K = W^k T; V = W^v T; F_{\text{bapf}} = LN(T + LN(softmax(\frac{QK^T}{\sqrt{d_k}})V)) \quad (2)$$

where $W$ is the weight, Q, K and V are the matrices after linear mapping of input, and $dk$ is the sequence length which is used to transform the attention matrix into a standard normal distribution.

To help regression transformers better learn and perceive the correlation between parts and bodies, we propose a reference plane consistency loss. We select the shoulders, hips and pelvis points on the human torso to establish the frontal plane of the torso, the side plane of the torso and the cross-section of the torso respectively (detailed diagram can be found in supplementary materials). In order to correspond to the 24 parts in the AGE module, we input the label parameters into the SMPL model to generate $gt\_vertices \in \mathbb{R}^{6890 \times 3}$, and take the center of each part area as the joint point. Define the joint point to bring into the plane equation greater than 0 as the positive direction, and calculate the depth value according to the distance from the point to the plane. Use this as the ground-truth label for relative depth, and construct explicit planar consistency constraints. This constraint helps the regression transformer to learn the spatial information between parts and bodies to solve the problem of depth ambiguity. Next, we take the output of the body-awareness encoder as the input to a single-layer regression transformer to learn dependencies among the features of each body-aware part. The final BAR module outputs a set of SMPL model parameters, camera parameters, and relative depth $RD \in \mathbb{R}^{24 \times 3}$ to the torso plane. We use the standard loss and the reference plane consistency loss for each part for training the network.

## 3.3 LOSS FUNCTIONS

We train ReBaR as a supervised learning problem, aiming to minimize the discrepancy between predicted outputs and ground truth annotations. To achieve this, we define a set of loss functions that capture different aspects of the model's performance. Given a training dataset $D$ which containing $N$ images, we have various ground truth data, including RGB images $I \in \mathbb{R}^{w \times h \times 3}$,

SMPL parameters $(\hat{\theta} \in \mathbb{R}^{24\times3}, \hat{\beta} \in \mathbb{R}^{10})$, relative depths $\hat{RD} \in \mathbb{R}^{24\times3}$, 3D and 2D coordinates $(\hat{J_{3D}} \in \mathbb{R}^{J\times4}, \hat{J_{2D}} \in \mathbb{R}^{J\times3})$ of body joints $J$, and segmentation labels for whole bodies and body parts $\hat{Seg}_{\text{parts}} \in \mathbb{R}^{H\times W\times24}$ and $\hat{Seg}_{\text{body}} \in \mathbb{R}^{H\times W\times1}$.

To supervise body and parts attention, we project the vertices generated by the SMPL model onto pixel coordinates, creating segmentation labels $Seg_{\text{body}} \in \mathbb{R}^{H\times W\times1}$ and $Seg_{\text{parts}} \in \mathbb{R}^{H\times W\times24}$. This allows us to supervise attention at the pixel level, providing more detailed information for evaluating model performance.

We use pose parameters $\theta$ and shape parameters $\beta$ to output body vertices $V_{3d} \in \mathbb{R}^{6890\times3}$. To compute the keypoint loss, we need the SMPL 3D joints $J_{3D}(\theta, \beta) = WV_{3d}$, which are computed using a pretrained linear regressor $W$. With the inferred weak-perspective camera, we compute the 2D projection of the 3D joints $J_{3D}$, as $J_{2D} \in \mathbb{R}^{J\times2} = s\Pi(RJ_{3D}) + t$, where $R \in SO(3)$ is the camera rotation matrix and $\Pi$ is the orthographic projection. We use the combination of loss functions to train ReBaR, including AGE loss and BAR loss, where each term is calculated as:

$$L_{\text{AGE}} = L_{\text{Aux2D}} + L_{\text{Aux3D}} + (L_{\text{BS}} + L_{\text{PS}}); L_{\text{BAR}} = L_{\text{2D}} + L_{\text{3D}} + L_{\text{SMPL}} + L_{\text{RD}}$$

$$L_{\text{joints}} = ||J_{\text{joints}} - \hat{J_{\text{joints}}}||_F^2; L_{\text{SMPL}} = ||\Theta - \hat{\Theta}||_2^2; L_{\text{RD}} = ||\text{RD} - \hat{\text{RD}}||_2^2$$

$$L_{\text{BS/PS}} = \frac{1}{\text{HW}} \sum_{h,w} CrossEntropy(\sigma(\text{ATT}_{h,w}), \text{Seg}_{h,w})$$

$$(3)$$

where $\hat{X}$ represents the ground truth for the corresponding variable $X$ and $L_{\text{joints}}$ is the supervision of all $3D$ and $2D$ keypoints. In addition, auxiliary supervision losses are $L_{\text{Aux2D}}$, $L_{\text{Aux3D}}$, $L_{\text{BS}}$ and $L_{\text{PS}}$. We train the model with all losses for the first 100 epochs. After that, we remove the segmentation supervision loss and continue training the model. This initial training phase helps the model learn to recognize and emphasize relevant regions in the input images. Once the attention mechanism has been guided towards the body parts, we remove the segmentation supervision loss and continue training. During this phase, the attention mechanism adapts further to better recover the human mesh from the images.

## 4 EXPERIMENTS

In our implementation, the input size of the network is $224\times224$. We use standard data augmentation practices during training, including random rotations, scaling, horizontal flipping, and cropping. Also, we adopt the widely used Adam optimizer to train the network for 240K steps with a batch size of 64 and a learning rate of $5 \times e^{-5}$.

**Datasets.** For the experiments on the 3DPW dataset, we used four large-scale datasets for training, including COCO-EFT (74K) (Lin et al., 2014), MPII (14K) (Andriluka et al., 2014), MPI-INF-3DHP (90K) (Mehta et al., 2017), and Human3.6M (292K) (Ionescu et al., 2013). We first pre-trained the network on COCO-EFT (74K) and then sequentially fine-tuned it on the mixed and 3DPW datasets. Next, we further fine-tuned the model on AGORA (Patel et al., 2021) and evaluated its performance on the AGORA test set.

**Comparison to the state-of-the-art.** We first comprehensively compare ReBaR with two types of state-of-the-art methods, including model-free and model-based methods on the 3DPW dataset. In this experiment, we report two evaluation results, namely 3DPW-ALL and 3DPW-TEST. Here, all methods do not include the 3DPW training dataset in training to evaluate the 3DPW-ALL setting, while for the 3DPW-TEST, they all fine-tune the network on the 3DPW training dataset. In addition, we provide the results of two network structures (HR-W48 and HR-W32) on these two settings, respectively. Table 2 presents the comparison results with various methods. Our method's performance using the HR-W32 backbone network has surpassed almost all existing methods in all metrics but is only slightly lower MVE than the CLIFF method using the HR-W48 backbone network. However, when using the HR-W48 backbone network, our method significantly outperforms the previous state-of-the-art method CLIFF on 3DPW-TEST, and MJE is reduced from 69.0 to 67.2. It is also worth mentioning that the evaluation results in 3DPW-ALL can better reflect the generalization ability of the method. Yet, our method also outperforms the previous state-of-the-art method HybrIK by a large margin (MJE from 80.0 to 72.0) in this setting.

We also evaluate our method on Human3.6M and multi-person dataset AGORA and compare it with state-of-the-art methods. Furthermore, since camera parameter estimation and kids model play

| Training | Method | AGORA | | Human3.6M | |
|---|---|---|---|---|---|
| | | MJE↓ | V2V↓ | MJE↓ | PAMJE↓ |
| Unfinetuned | SPIN (Kolotouros et al., 2019) | 175.1 | 168.7 | - | - |
| | PyMAF (Zhang et al., 2021) | 174.2 | 168.2 | - | - |
| | EFT (Joo et al., 2021) | 165.4 | 159.0 | - | - |
| | PARE (Kocabas et al., 2021a) | 146.2 | 140.9 | - | - |
| | **ReBaR (Ours)** | **134.6** | **128.9** | - | - |
| Finetuned | ROMP (Sun et al., 2021) | 108.1 | 103.4 | - | - |
| | BEV (Sun et al., 2022) | 105.3 | 100.7 | - | - |
| | FastMETRO–L–H64 (Cho et al., 2022) | - | - | 52.2 | 33.7 |
| | VisDB (Yao et al., 2022) | - | - | 51.0 | 34.5 |
| | Hand4Whole (Moon et al., 2022) | 89.8 | 84.8 | - | - |
| | CLIFF (Li et al., 2022) | 81.0 | 76.0 | 47.1 | 32.7 |
| | **ReBaR (Ours)** | **79.9** | **74.5** | **45.4** | **32.1** |

Table 1: **Performance comparison on the AGORA and Human3.6m dataset.** In the upper segment, we present results from a model without fine-tuning on the AGORA dataset. In the lower portion, we showcase the refined outcomes after the fine-tuning process.

| Method | 3DPW-ALL | | | 3DPW-TEST | | |
|---|---|---|---|---|---|---|
| | MJE↓ | PAMJE↓ | MVE↓ | MJE↓ | PAMJE↓ | MVE↓ |
| I2L-MeshNet (Moon & Lee, 2020) | 93.2 | 58.6 | - | - | - | - |
| METRO (Lin et al., 2021) | - | - | - | 77.1 | 47.9 | 88.2 |
| HybrIK (Li et al., 2021) | 80.0 | 48.8 | 94.5 | 74.1 | 45.0 | 86.5 |
| FastMETRO–L–H64 (Cho et al., 2022) | - | - | - | 73.5 | 44.6 | 84.1 |
| HMR (Joo et al., 2020) | 85.1 | 52.2 | 118.5 | - | - | - |
| ROMP (Sun et al., 2021) | 82.7 | 60.5 | - | 76.7 | 47.3 | 93.4 |
| PARE (Kocabas et al., 2021a) | 82.0 | 50.9 | 97.9 | 74.5 | 46.5 | 88.6 |
| VisDB (Yao et al., 2022) | - | - | - | 72.1 | 44.1 | 83.5 |
| CLIFF(HR-W48) (Li et al., 2022) | - | - | - | 69.0 | 43.0 | 81.2 |
| MotionBERT (HR-W48) (Zhu et al., 2022) | - | - | - | 68.8 | **40.6** | 79.4 |
| ProPose (HR-W48) (Fang et al., 2023) | - | - | - | 68.3 | **40.6** | 79.4 |
| ReBaR (HR-W32) | 76.1 | 45.9 | 90.9 | 69.1 | 41.8 | 81.9 |
| ReBaR (HR-W48) | **72.0** | **45.3** | **86.5** | **67.2** | 40.8 | **78.7** |

Table 2: **Performance comparison on the 3DPW dataset.** For the evaluation on 3DPW-ALL, all methods are trained without 3DPW datasets.

a crucial role in the performance of the AGORA dataset, we compare our method with those without kids model and using weak-perspective projection in training for a fair comparison. Table 1 shows the result where our method still performs better than multi-person-based approaches and previous state-of-the-art method CLIFF.

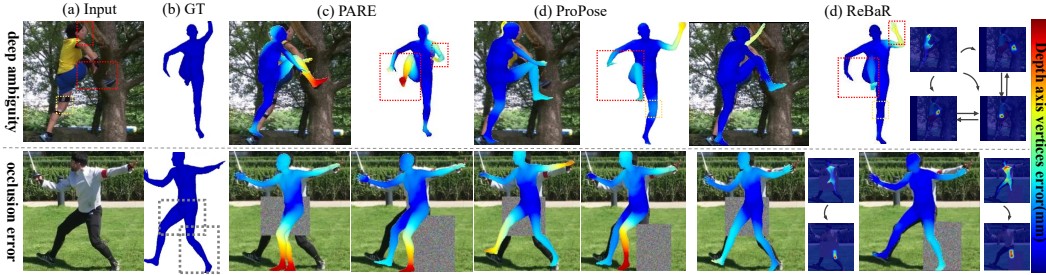

Figure 3: **Occlusion and Depth error processing.** Compared to PARE and ProPose, our ReBaR exhibits enhanced prediction performance for occluded body parts. From a side view perspective, ReBaR is noticeably superior to PARE in terms of accuracy along the depth axis.

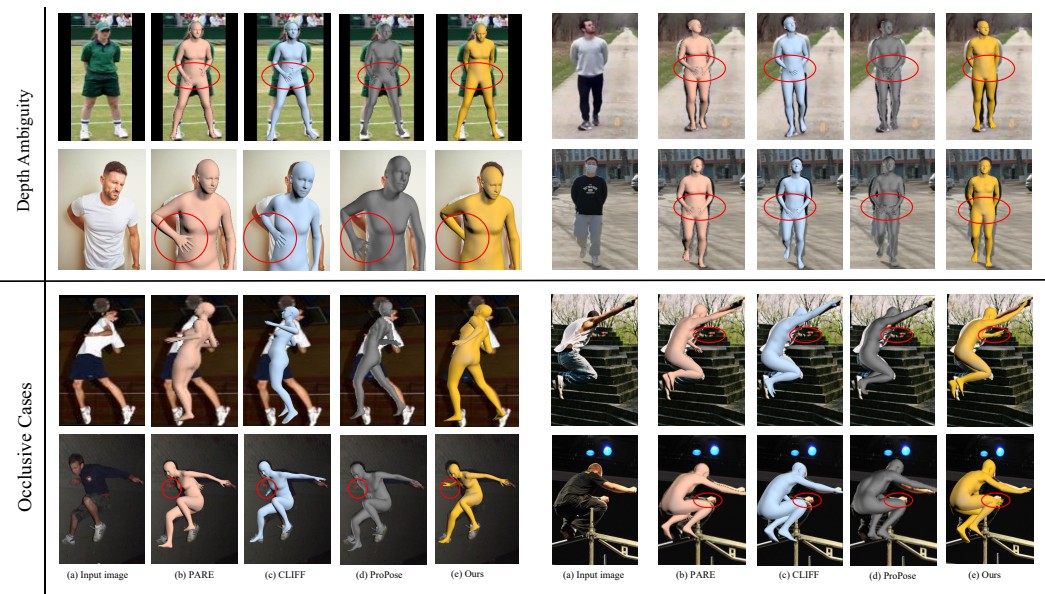

Figure 4: **Qualitative comparison of PARE, CLIFF, and our method ReBaR in occluded and depth-ambiguous scenarios.**

**Improved occlusion handling.** Table 3 showcases the results of our proposed ReBaR method, the baseline PARE method, and the CLIFF method on the occlusion datasets 3DPW-OCC and 3DOH. All methods were evaluated on 3DPW-OCC without using the 3DPW training dataset. When evaluating 3DOH, they fine-tuned the network on the 3DPW training dataset. Both evaluations excluded their own training set to ensure a better comparison of the generalization ability of the methods on occlusion datasets. ReBaR outperformed PARE in both MJE and PAMJE, and it also showed superior performance compared to CLIFF and PARE on 3DOH. Specifically, on the 3DPW-OCC dataset, ReBaR reduced MJE and PAMJE by 5.8% and 10.9%, respectively, compared to PARE. On 3DOH, ReBaR achieved a 2.6% improvement over CLIFF. These results and Figure 3 suggest that part-aware regression with body reference adjustment enhances occlusion handling and alleviates depth ambiguity, achieving improved accuracy and generalization in challenging scenarios.

| Method | 3DPW-OCC | | 3DOH | | 3DPW-TEST | | |
|---|---|---|---|---|---|---|---|
| | MJE↓ | PAMJE↓ | MJE↓ | PAMJE↓ | $MJE_x$ | $MJE_y$ | $MJE_z$ |
| PARE (HR-W32) | 84.9 | 57.5 | 94.9 | 62.4 | 30.3 | 28.9 | 58.2 |
| CLIFF (HR-W48) | - | - | 80.4 | 58.7 | 27.0 | 27.1 | 51.8 |
| ReBaR (HR-W32) | 81.7 | 49.5 | 82.7 | **55.5** | 25.9 | 27.5 | 46.6 |
| ReBaR (HR-W48) | **74.6** | **48.3** | **78.3** | 56.3 | **23.6** | **25.0** | **45.5** |

Table 3: **Comparison along the depth axis on 3DPW and OCC datasets.** The main improvement of ReBaR compared to CLIFF and PARE lies in its significant enhancement along the depth axis (Z-axis).

**Reduced depth ambiguity.** To further analyze the issue of depth ambiguity in 3D human reconstruction, we evaluate the average per joint position error (MJE) in the x, y, and z axes on the 3DPW-Test dataset. As shown in Table 3 and Figure 3, we can see that the error on the z-axis is significantly higher than that on the x and y axes in existing methods. This indicates that the depth ambiguity problem in 3D human reconstruction is a significant challenge that needs to be addressed. However, our approach outperforms the state-of-the-art methods by a significant margin in the z-axis metric, achieving an 10% improvement compared to CLIFF. This result highlights the importance of body-aware part features encoding, which enables the inference of spatial relationships between body and parts through local visual clues around body parts and relevant global information from

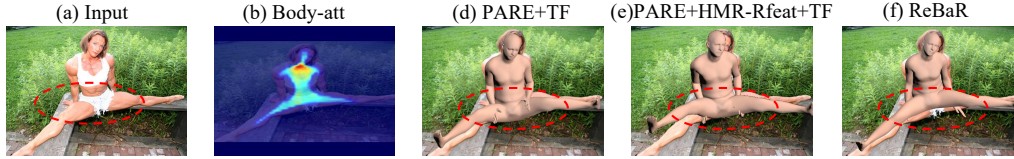

Figure 5: Qualitative analysis of the effectiveness of Attention-guided reference features (Att-RFeat).

the whole body, effectively alleviating the problem of depth ambiguity.

**Qualitative comparison.** In Figure 4, we compare the performance of PARE, CLIFF, and ReBaR on different test datasets. To render the human geometry created by the SMPL model into images, we use the predicted camera parameters and weak perspective projection (Kissos et al., 2020). The results show that PARE and CLIFF struggle to accurately infer the depth information between limbs and exhibit ambiguities in the orientation of the human body under large occlusions. However, thanks to the body reference condition dependency established by the BAR module, ReBaR can more accurately recover these motions.

**Ablation Experiments.** In this experiment, we conducted ablation studies on the primary modules of our method, with more detailed ablation available in the appendix. All compared models use the HR-W32 backbone network and are trained on mixed data. Table 4 presents our experimental results. As seen from the table, directly combining the Pare method with the Transformer results in only a slight improvement in Pare (MJE reduced from 74.5 to 73.2). However, upon incorporating our Attention-guided reference features (**ATT-RFeat**), the overall performance significantly improves (MJE reduced from 74.5 to 70.4, and MJE Depth reduced from 58.2 to 47.5). This confirms that our **ATT-RFeat** is the primary contributor to the performance enhancement and effectively addresses deep ambiguity. Furthermore, the results demonstrate that adding relative distance loss further boosts overall performance. We also verified that the Transformer (**TF**) outperforms the **MLP**. While using the HMR feature as a reference feature (**HMR-RFeat**) can improve baseline performance, our **ATT-RFeat** is superior to the **HMR-RFeat**. At the same time, our qualitative results in Figure 5 also prove that our method more effectively handles occlusion and depth ambiguity than **HMR-RFeat**.

| Method | MJE | PAMJE | MVE | $MJE_z$ |
|---|---|---|---|---|
| Baseline (PARE) | 74.5 | 46.5 | 88.6 | 58.2 |
| Baseline+TF | 73.2 | 44.7 | 85.0 | 56.6 |
| Baseline+HMR-RFeat+MLP | 72.7 | 43.9 | 82.7 | 53.1 |
| Baseline+HMR-RFeat+TF | 71.5 | 43.3 | 83.8 | 49.0 |
| Baseline+ATT-RFeat+TF | 70.4 | 42.4 | 82.2 | 47.5 |
| Baseline+ATT-RFeat+TF+$L_{RD}$ (ReBaR) | 69.1 | 41.8 | 81.9 | 46.6 |

Table 4: Ablation studies on the primary modules of our method.

## 5 CONCLUSION

In conclusion, our proposed body-aware part regressor significantly improves human body shape and pose estimation from monocular images compared to current methods. By utilizing a soft attention mechanism to capture global information and dependencies between body parts, and implementing body-aware regression using both local and global information, our method handles severe occlusions and depth ambiguity, achieving state-of-the-art performance across various datasets and scenarios. This work advances human body shape and pose estimation and offers valuable insights for future research.. Nonetheless, our method faces challenges with extreme perspectives, causing imprecise torso representations and impacting performance. Future work will explore adaptive reference feature selection.

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

## A  APPENDIX

This supplement provides additional experimental results to enhance the main paper. In Section B, we offer more implementation details. In Section C, we present additional experimental results. In Section D, we use more in-the-wild data to compare and verify the effectiveness of ReBaR on extremely challenging real-world problems.

## B  MORE DETAILS

In our experiments, our backbone network is initialized by the HRNet model weights pre-trained on the MPII (Andriluka et al., 2014) dataset for 2D key point detection tasks.

**Torso Plane and Relative Depth**
As shown in figure 7, we select the shoulder, hip, and pelvis points on the human torso to establish the frontal, lateral, and transverse planes of the torso. The center points of the skin regions of each joint are defined as the actual joint positions, and joint axes are defined by bringing equations greater than 0 into the plane as positive directions, and the depth value is calculated based on the distance from the point to the plane. This is used as a basic fact label for relative depth, and explicit plane consistency constraints are constructed.

**Loss Weight**
The weight distribution of our supervision functions is as follows: segmentation map supervision, reference feature 2D/3D auxiliary constraints, and relative depth supervision all have a weight of 60, while the SMPL parameter and keypoint supervision are five times greater than the former. This balanced approach contributes to the model's effectiveness.

**Data Augmentation**
In all the experiments in the paper, except that Table 2 does not use any augmentation on the AGORA dataset, other experiments use the same data augmentation method as PARE.

**Auxiliary Loss**
In the process of training ReBaR, we respectively use global 2D/3D keypoints, body/part attention map and relative depth supervision to assist the network to learn body-aware part features. For the supervision of body/part attention, we only supervise on the COCO-EFT dataset, and reset the weight of $L_{x\_seg}$ in the loss function to 0 on the mix and 3DPW datasets.For the AGORA dataset, we do not supervise the relative depth.

**Inference Time and parameter Counts**
Our model employs a Transformer to trade computational cost for exceptional performance, resulting in a longer inference time compared to PARE and CLIFF, but still within an acceptable range. As shown in Table 5, we tested the inference speed for a single image using the same GPU, T4-8C, and calculated the total number of parameters for the PARE, CLIFF, and ReBaR models. Additionally, we recorded the model training costs in the log files, with training conducted on four V100 GPUs. PARE takes 16 hours to train with the original settings (72 hours on a 2080Ti), while ReBaR requires 8 hours for training on COCO and 20 hours for fine-tuning on the mixed dataset. Lastly, when evaluating AGORA, we fine-tuned the model for 10 hours on a single V100 GPU using the training set. CLIFF does not provide open-source training code, so it is not included in the statistics.

**Part/Body segmentation maps**
To obtain segmentation maps for the auxiliary supervision of the body attention map and part attention map in the AGE module, segmentation labels are required for each part of the original image. However, labeling the original image is a time-consuming and costly process. Luckily, we can utilize the SMPL model to obtain the vertex coordinates in the camera coordinate system that correspond to each image. Using weak perspective transformation techniques (Kissos et al., 2020), we can then generate the 2D vertex coordinates in the pixel coordinate system that are needed to generate the segmentation map. This method enables us to obtain the necessary segmentation maps without the need for costly and manual labeling of the original image. We introduce inverse weak-perspective projection to generate body/part attention map labels. First, let's discuss weak-perspective projection:

| Method | PARAMS | Inference | Training | FLOPS | $MJE$ | $MJE_z$ |
|---|---|---|---|---|---|---|
| PARE | 125.5MB | 1.24s | 16h | 14.9G | 74.5 | 58.2 |
| PARE + TF | 155.8MB | 1.32s | 19h | 18.1G | 73.2 | 56.6 |
| CLIFF* | 305MB | 1.52s | - | - | 69.0 | 51.8 |
| ReBaR (Ours) | 222MB | 1.47s | 28h | 26.4G | 69.1 | 46.6 |
| ReBaR* (Ours) | 361.1MB | 1.81s | 45h | - | 67.2 | 45.5 |

Table 5: **More details about the model.** * means using HRNet-48.

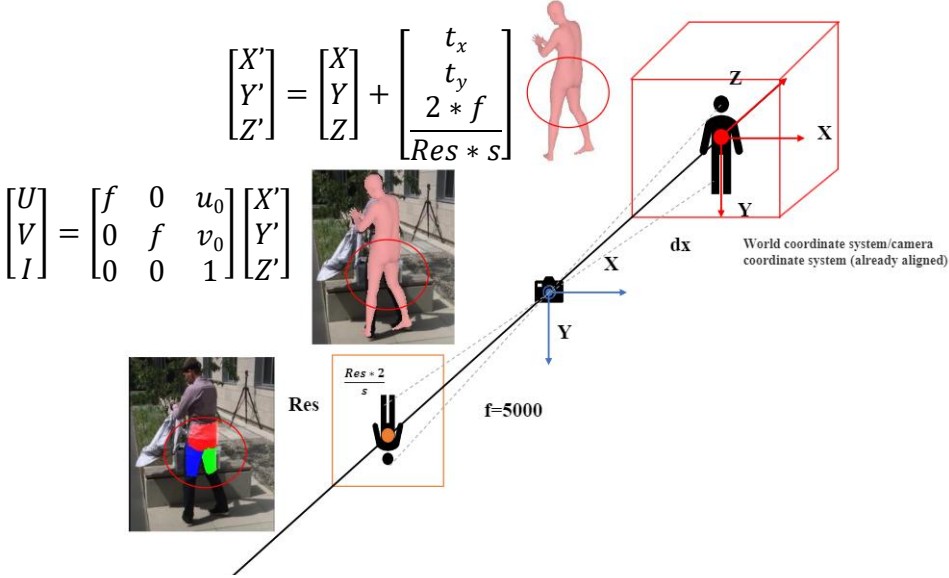

Figure 6: **Weak-perspective projection illustration.**

The weak-perspective projection, as described in Kissos et al. (2020), is based on the premise that the focal length and object distance are sufficiently large, allowing for the neglect of variations in the object along the Z-axis. It is a technique for converting three-dimensional camera coordinates into pixel coordinates. Before performing the projection, the 3D keypoints are normalized to a cube in the range [-1, 1], and the camera is aligned to the world coordinate system origin (the cube's center). Next, the projection camera parameters $(s, t_x, t_y)$ are required, where s represents the human body's scale in the cropped image, and $t_x$ and $t_y$ represent the translation within the cube. In this study, the cropping size Res is 224. To perform the weak-perspective projection, we first increase the fearlessness, as shown in the following formula:

$$t_z = \frac{2 \times f}{Res \times s} \tag{4}$$

where, f represents the focal length, which is set to 5000 in our study. Subsequently, we add $t_x$, $t_y$, and $t_z$ as translations to the 3D keypoints and introduce the following formula for weak-perspective projection:

$$\begin{bmatrix} U \\ V \\ I \end{bmatrix} = \begin{bmatrix} f & 0 & u_0 \\ 0 & f & v_0 \\ 0 & 0 & 1 \end{bmatrix} \begin{bmatrix} X + t_x \\ Y + t_y \\ Z + t_z \end{bmatrix} \tag{5}$$

where we set $u_0$ and $v_0$ to the image's center point position, and f represents the focal length. The resulting $U$ and $V$ are the 2D keypoints in the pixel coordinate system after projection. In the reverse process, we obtain 3D mesh points and 3D/2D keypoints using the ground truth (GT) labels and the SMPL model. We then compute the translation in reverse using the 3D/2D keypoints. Subsequently, we add the translation to the 3D mesh points in the same manner and perform weak-perspective projection to obtain body/part attention map labels. The process is shown in Figure 6.

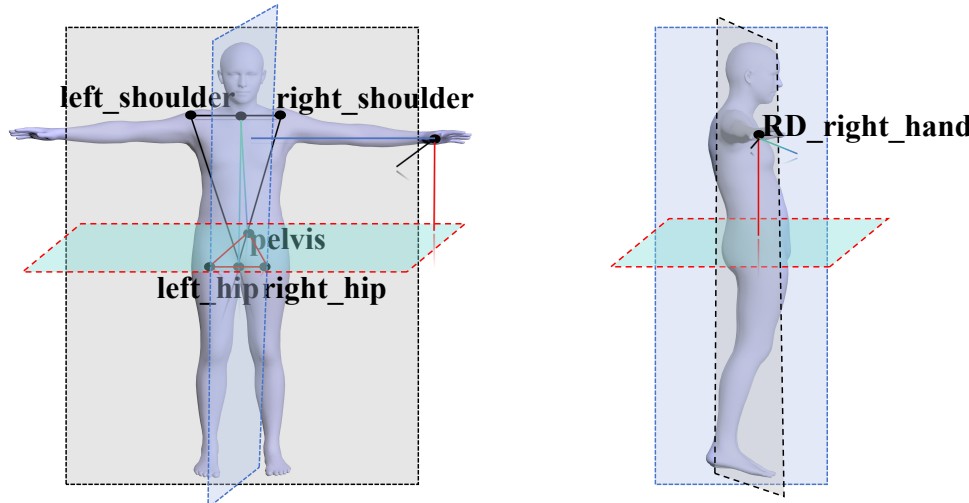

Figure 7: **Illustration of the calculation of the relative depth from the torso plane.** Shows the torso points for constructing the torso planes and the relative depth of the right hand.

## C  EXPERIMENTS

In this section, we supplement more experimental results and ablation experiments to verify the effectiveness of ReBaR.

| Method | Elbow | | | Wrist | | | Head | | |
|---|---|---|---|---|---|---|---|---|---|
| | MJE$_x$↓ | MJE$_y$↓ | MJE$_z$↓ | MJE$_x$↓ | MJE$_y$↓ | MJE$_z$↓ | MJE$_x$↓ | MJE$_y$↓ | MJE$_z$↓ |
| PARE | 33.7 | **27.5** | 55.0 | 42.6 | 36.1 | 80.5 | 29.5 | 33.9 | 70.6 |
| ReBaR | **27.9** | 27.9 | **48.5** | **36.1** | **35.5** | **75.6** | **20.5** | **30.0** | **48.1** |

| Method | Ankle | | | Knee | | | Neck | | |
|---|---|---|---|---|---|---|---|---|---|
| | MJE$_x$↓ | MJE$_y$↓ | MJE$_z$↓ | MJE$_x$↓ | MJE$_y$↓ | MJE$_z$↓ | MJE$_x$↓ | MJE$_y$↓ | MJE$_z$↓ |
| PARE | 46.3 | 47.8 | 82.2 | 27.7 | 26.7 | 47.5 | 18.2 | 27.3 | 46.8 |
| ReBaR | **35.0** | **36.6** | **71.0** | **21.8** | **17.9** | **40.4** | **15.3** | **24.1** | **36.0** |

Table 6: **Part-by-part performance comparison on the 3DPW-Test.** All methods have been trained on dataset with 3DPW.

**Per-part Evaluation**
To validate the effectiveness of our method, we conducted more comprehensive evaluation experiments and reported the MPJPE metrics for each body part on each axis in Table 6. Compared to PARE, our method achieved a significant improvement of about 8mm on most parts, while the improvement of the shoulder in the z-axis is relatively small. Notably, our method achieved the most significant improvement on the Head, which dropped by 23.5mm in the z-axis. Some body parts such as the ankle and neck also showed substantial improvement, with a drop of 11.2mm and 10.8mm in the z-axis, respectively.

**Analysis of BAR Module**
BAR is the core part of ReBaR. Its role is to construct body perception component features and establish connections between components, so as to infer the posture of occluded parts through more reasonable visible information. This also makes ReBaR have a better ability to alleviate the depth blur problem than existing methods. As shown in Figure 8, we directly associate the component features of PARE and cannot correctly infer the correct posture of the occluded forearm (such as the wrong posture of the forearm bending forward), but after adding the body reference condition, the model infers Relatively correct posture, although its depth information is not very accurate, and then with the help of relative torso depth constraints, ReBaR accurately infers the posture of the arms behind it. In addition, we also found that in some extremely challenging actions, the body reference condition can greatly improve the stability of the root node. As shown in Figure 8, in the inverted

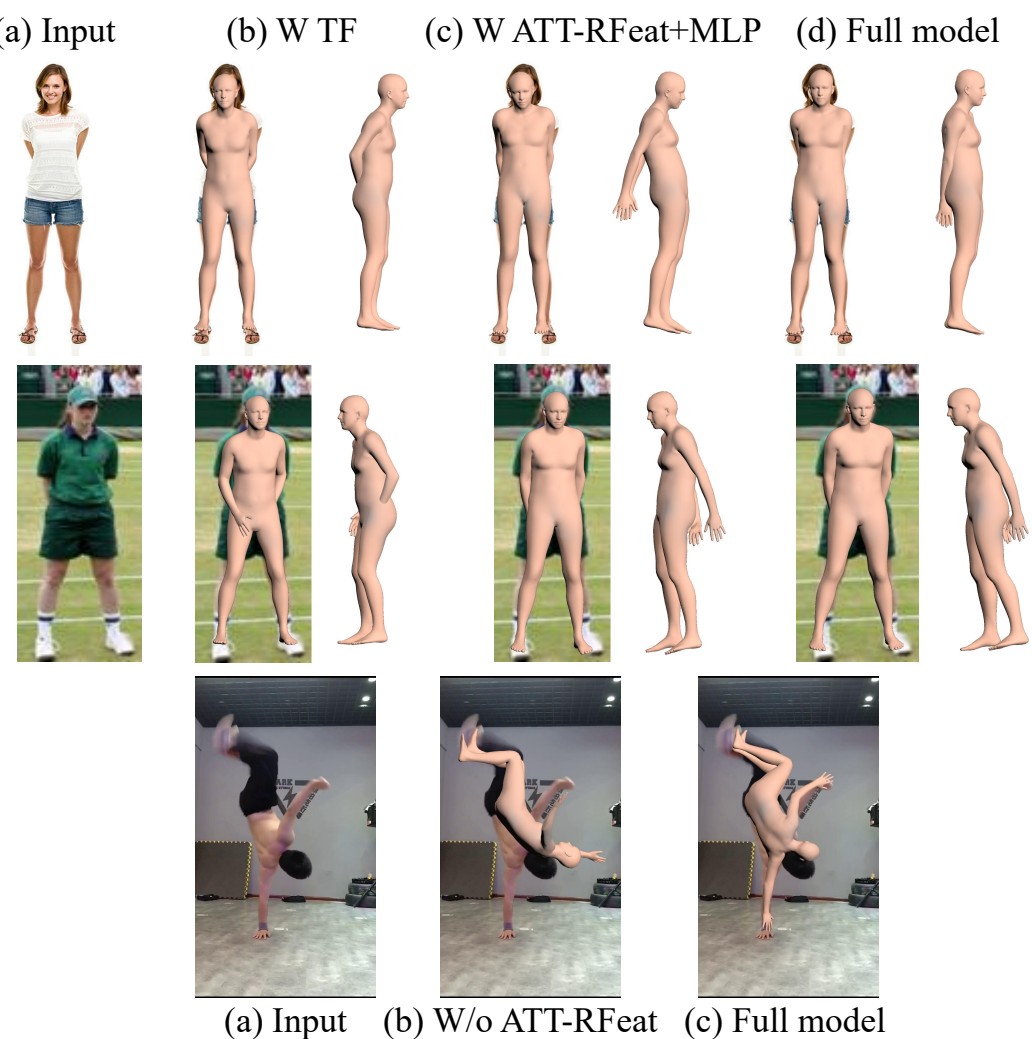

(a) Input (b) W TF (c) W ATT-RFeat+MLP (d) Full model

(a) Input (b) W/o ATT-RFeat (c) Full model

Figure 8: **The role of BAR module and ATT-RFeat.** For the first two rows, from left to right: input image, PARE+transformer, PARE+BAR, and the result of the full model. For the last line, from left to right: input image, w/o ATT-Rfeat, and the result of the full model.

| Method | 3DPW-All | | |
| --- | --- | --- | --- |
| | MJE↓ | PAMJE↓ | V2V↓ |
| Baseline (Kocabas et al., 2021a) | 91.0 | 56.7 | 108.2 |
| Baseline w TF | 91.5 | 55.7 | 108.6 |
| Baseline w HMR-RFeat+TF | 90.1 | 55.0 | 106.1 |
| ReBaR w/o $L_{3D}$ | 89.4 | 55.8 | 106.7 |
| ReBaR w/o $L_{2D}$ | 90.6 | 54.8 | 107.2 |
| ReBaR w/o $L_{RD}$ | 89.0 | 54.2 | 105.7 |
| ReBaR | **88.6** | **53.7** | **105.3** |

Table 7: **Ablation study of ReBaR on 3DPW.** All methods are trained on COCO-EFT-PART.

action, because PARE independently predicts the root joint, it causes an error in the global rotation direction. However, our method ReBaR accurately predicts a reasonable direction through the root feature of body perception, avoiding the problem of random rotation of the human body in video capture.

**The Role of Attention-guided Reference Feature**
Figure 9 illustrates the qualitative improvement of our method in challenging cases such as severe occlusion, challenging poses, and depth ambiguity. This demonstrates the importance of body-aware part features encoding and utilizing visual information around parts and Attention-guided reference feature to address depth ambiguity and self-occlusion issues.

**Ablation Experiments With Attention-guided Reference Features**
To evaluate the effectiveness of the body-aware regressor module, we conducted a set of comparative experiments. We used the global feature of the HMR model as a reference and directly associated it with the part feature to regress the SMPL model parameters using the same transformers. The results, as shown in Table 7, indicate that the global reference encoding only marginally improves the PAMJE metric compared to PARE. However, when we replaced the HMR feature with the Attention-guided reference feature of the AGE module, the model's performance significantly improved. This demonstrates the importance of using Attention-guided referenced feature for inferring part poses.

**Ablation Experiments With Auxiliary Constraints**
In this experiment, we use HR-W32 as the backbone and train all comparison methods on the small COCO-2014-EFT (22K) dataset (a subset of the COCO-EFT dataset). We first combine PARE with Transformers and evaluate the performance on the 3DPW dataset. Table 7 shows that integrating both techniques slightly reduces PAMJE but increases MJE and PVE. In contrast, our proposed Re-BaR significantly improves PARE, indicating that learning Attention-guided reference feature substantially contributes to the performance gain. Furthermore, we validate the auxiliary constraints, i.e., 2D keypoints $L_{2D}$, 3D keypoints $L_{3D}$, and relative depth $L_{RD}$, which bring different levels of improvement to our proposed Attention-guided reference feature. Compared to the unconstrained HMR-feature and single-information supervised body-feature, establishing dependencies between 2D and 3D information in space can better construct stable reference conditions, thereby greatly improving joint prediction accuracy. The relative depth constraint provides greater weight on the depth axis and endows the model with the ability to perceive front-back relationships (positive outside the torso plane and negative otherwise), thereby alleviating the depth blur problem, which is an ability that pure 3D keypoint constraints do not possess. The results in Table 7 show that all losses contribute to improved performance.

**Compared To Video-based Methods**
We also compared our method to the state-of-the-art video-based methods. Table 8 shows the results, which demonstrate that our method outperforms these methods by a significant margin even without additional temporal information from the video.

## D    MORE QUALITATIVE COMPARISONS

In this section, we provide more qualitative comparison results. Visualize PARE (Kocabas et al., 2021a), CLIFF (Li et al., 2022) and ReBaR results on images and challenging video files respectively

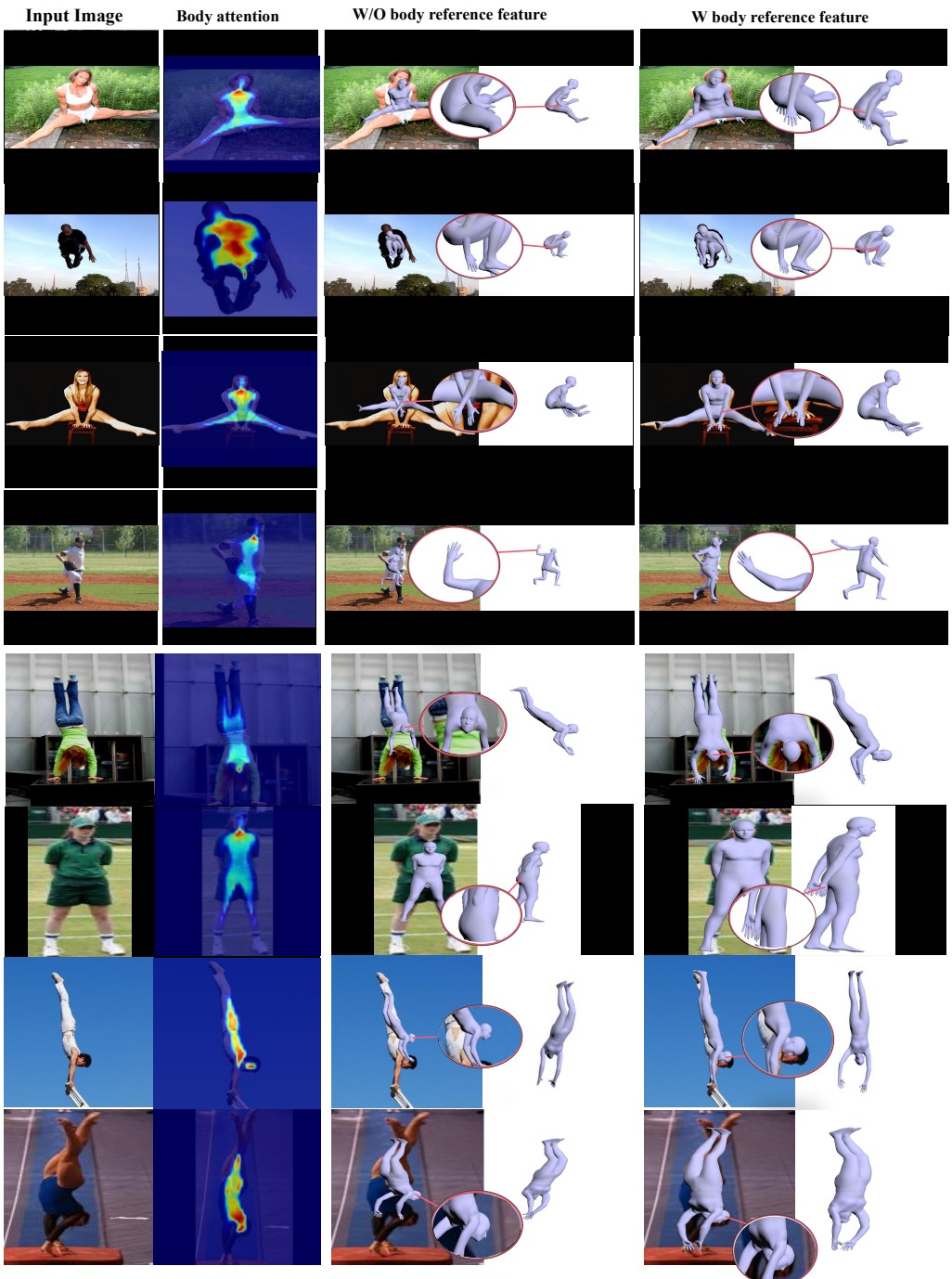

Figure 9: **The role of Attention-guided reference feature.** From left to right: input image, body attention map, the result of discarding Attention-guided reference feature, and the result of the full model.

| Method | 3DPW-Test | |
| --- | --- | --- |
| | MJE↓ | V2V↓ |
| HMMR (Kanazawa et al., 2018) | 116.5 | 72.6 |
| Doersch et al. (Doersch & Zisserman, 2019) | - | 74.7 |
| Sun et al. (Sun et al., 2019) | - | 69.5 |
| TCMR (Choi et al., 2021) | 105.3 | 100.7 |
| VIBE (Kocabas et al., 2020) | 82.7 | 51.9 |
| MAED (Wan et al., 2021) | 79.1 | 45.7 |
| ReBaR | **69.1** | **41.8** |

Table 8: **Evaluation on 3DPW-Test.** ReBaR achieves the best results without using the Euro filter."-" means no data provided.

for comparison. The image data includes two public datasets of 3DPW (Von Marcard et al., 2018) and LSPET (Johnson & Everingham, 2011), which contain problems such as occlusion, direction blur, and depth blur. Video files are downloaded directly from the internet and include challenging action sequences such as yoga, hip-hop and more.

**Qualitative Comparison of Image** As shown in Figure 10, our method outperforms PARE and CLIFF in almost all cases on the motion dataset LSPET. Especially under the problem of depth ambiguity and self-occlusion, thanks to the body-aware part features encoding, ReBaR can infer the spatial relationship between the occluded part and the body from the local visual cues around the part and the relevant global information of the whole body, Thereby improving the estimation accuracy. More interestingly, we found that PARE can barely determine the body orientation in some handstand situations, which we believe is due to PARE disconnecting the limbs and independently predicting the global rotation part. However, ReBaR avoids this problem nicely by using body reference conditions and limb dependencies.

**Qualitative Comparison of Challenging Videos**
As shown in Figure 11, we intercept some video frames for qualitative comparison. It is not difficult to find that in challenging action sequences such as hip-hop and yoga, the effects of PARE and CLIFF have dropped significantly, which shows that neither completely independent part prediction methods nor pure global prediction methods can handle actual complex movements in real-world applications. But ReBaR showed good results in handling these challenging actionse.

# E   LIMITATIONS AND FAILURE CASES

As shown in the Figure 12, we present a set of challenging scenarios where ReBaR struggles to achieve satisfactory reconstruction results. In these situations, the camera captures the subjects from either a bird's-eye or worm's-eye view, making the human torso nearly invisible. This factor makes it difficult to accurately extract our reference features, resulting in an imprecise human torso representation that negatively impacts the reconstruction performance. It is important to note that these challenging scenarios frequently appear in motion data, highlighting an ongoing issue that requires further investigation and improvement.

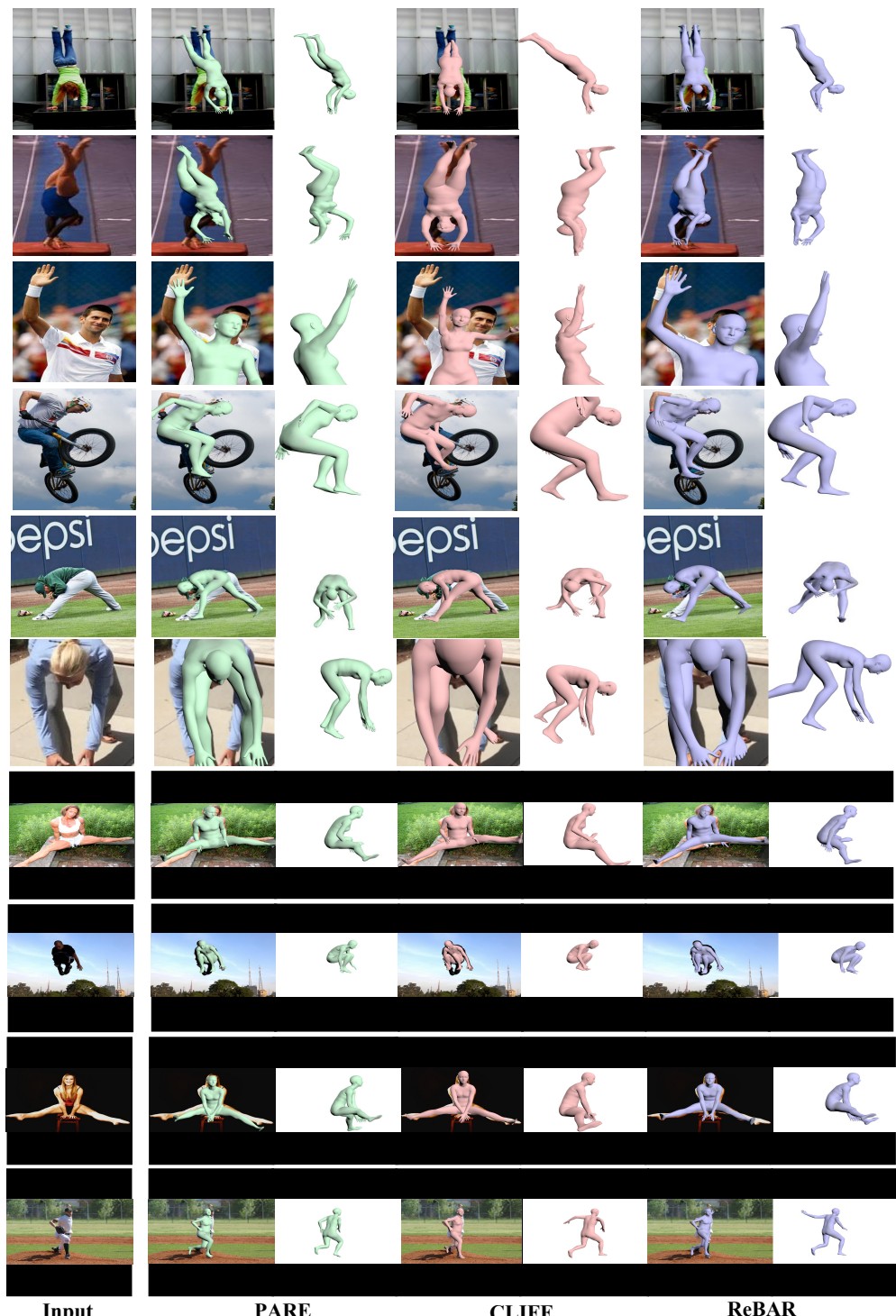

Figure 10: **Qualitative comparison of image.** From left to right: input image, PARE, CLIFF and ReBaR.

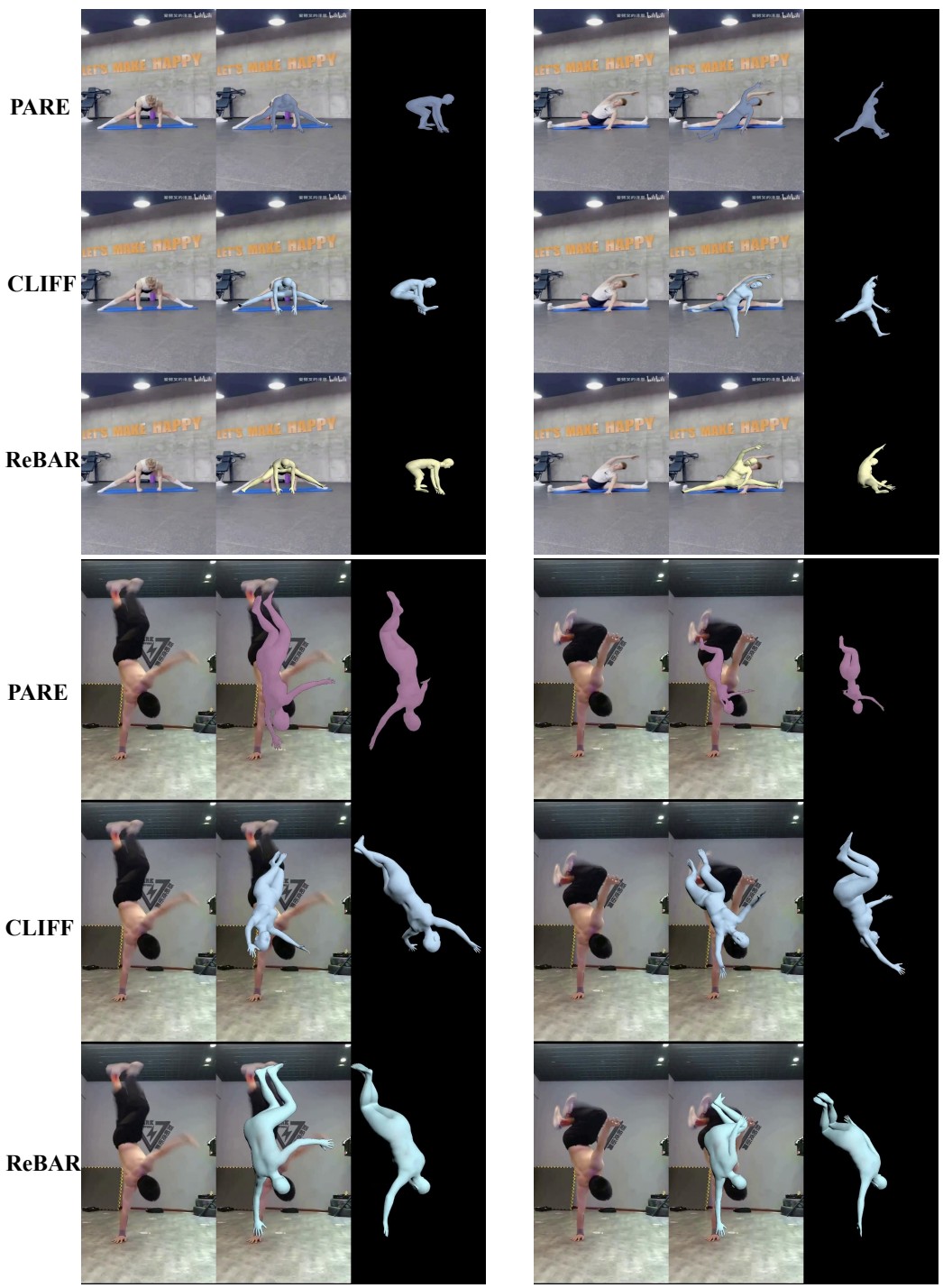

Figure 11: **Qualitative comparison of video.** From up to down: PARE, CLIFF and ReBaR.

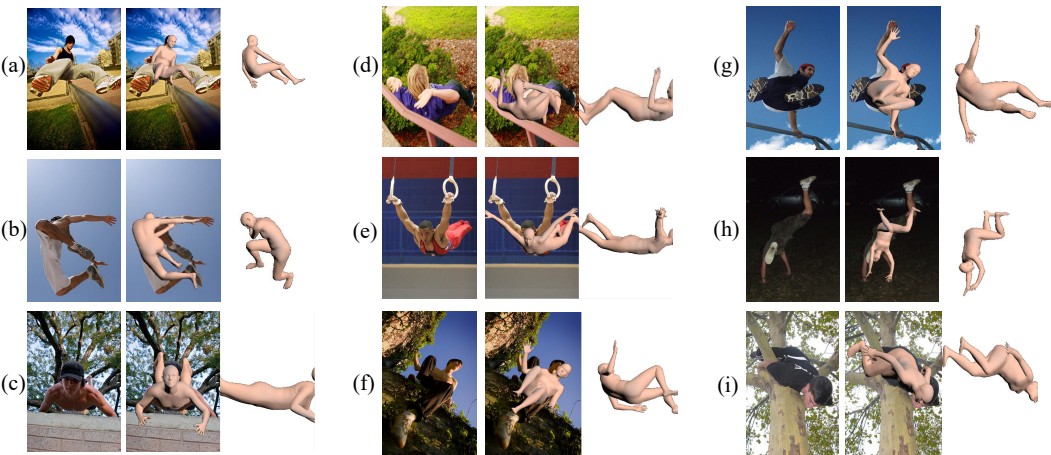

Figure 12: In highly challenging complex motion scenarios captured from bird's-eye or worm's-eye perspectives, ReBaR struggles to reconstruct satisfactory results when reference features are not well-defined.

