# OpenReview forum: "ReBaR: Reference-Based Reasoning for Robust Human Pose and Shape Estimation from Monocular Images"
_ICLR.cc/2024/Conference — Submitted to ICLR 2024_

### Official Review · Reviewer_VfGb · 2023-10-14

**Soundness:** 3 good
**Presentation:** 2 fair
**Contribution:** 2 fair
**Rating:** 5
**Confidence:** 5

**Summary:**

This paper presents a 3D human reconstruction system. PARE is used as a baseline, and there are several technical contributions to make it better, which include 1) introducing Transformer, 2) utilizing body features (global feature), and 3) relative depth estimation. Experimental results demonstrate the effectiveness of the proposed system.

**Strengths:**

Experiments are extensively conducted and demonstrate the effectiveness of the proposed system.

**Weaknesses:**

1. Technical contribution is limited. Table 4 clearly shows that there are noticeable performance improvements from the baseline, but what the authors did is not very novel actually. Utilizing the body feature (global feature) increases the performance a lot, which shows that using both local and global features is necessary. This makes sense, but is kind of expected.

2. I’m not sure why we need the relative depth loss function. The 3D joint angles already provide relative depth information. Why we need this?

3. Writings and notations are not clear. There are a couple of words that represent the same thing: reference feature and body feature. All of them are not very clear. Why not just call them a global feature, which is more intuitive and easy to get? Texts for description (for example, AGE in L_AGE) should be written with $L_\text{AGE}$. Also, the authors used $R$ to represent a set of real numbers, which should be $\mathbb{R}$.

**Questions:**

1. There are two ‘GAP’ in Figure 2. What does it mean? Does it mean global average pooling? The main manuscript says nothing about GAP.
2. Hand4Whole in Table 1 has wrong reference. It is from Moon et al (CVPRW 2022).
3. How is the part segmentation GT obtained? The authors rendered SMPL meshes by considering visibility as well? For example, the occluded body part has GT part segmentation?

---

> ### Author Response · Authors · 2023-11-22
> **Reply to Reviewer VfGb**
>
> We have taken your comments into consideration and made the following improvements:
> 1. Restructured the content in the method section for better organization and flow.
> 2. Revised the writing to improve clarity and conciseness.
> 3. Enhanced the quality and alignment of figures and tables.
> 4. Corrected any typos and grammatical errors.
> 5. Corrected the erroneous references.
> These improvements significantly enhance the overall quality and presentation of our paper, making it more accessible and informative for readers. We truly appreciate your time and effort devoted to reviewing and providing valuable, detailed feedback on our paper.
>
> $\textbf{Q1 Technical contribution is limited.}$ We appreciate your feedback on the technical contribution of our work. While it is true that the utilization of both local and global features is an expected approach, we would like to emphasize that our proposed method goes beyond merely combining these features. Our method introduces a novel body-reference attention mechanism and leverages the Transformer architecture to effectively process and integrate both local and global features, resulting in noticeable performance improvements over the baseline, as shown in $\textbf{Table $\color{red}{4}$}$.
> The attention mechanism helps our model focus on relevant body parts and reference region in the input images, while the Transformer architecture enables their dependencies. This combination of techniques contributes to the improved performance of our method.
> We acknowledge that the core idea of using both local and global features may not be groundbreaking; however, the way we have implemented and integrated these features in our model, along with the body-reference feature learning and Transformer architecture, represents a meaningful contribution to the field. We believe that our work provides valuable insights for handling occlusion and depth ambiguity by learning a reference feature .
>
> $\textbf{Q2 Why we need the relative depth loss function?}$ Thank you for your insightful question. It is true that the 3D joint angles already provide relative depth information. However, by defining three orthogonal planes and measuring the distance of a point to these planes, the relative depth loss captures the relative relationships between joints and the planes, which can further improve the performance due to the following possible reasons:
> 1. Local depth information: While regressing 3D GT labels primarily focuses on the global structure and pose of the human body, the relative depth loss emphasizes local depth information by considering the pairwise relationships between body joints and the planes. This helps the model to capture more subtle depth variations and improve the overall depth estimation.
> 2. Improved generalization: The relative depth loss  encourages the model to learn depth patterns that are more generalizable to different poses and body shapes. Since it focuses on the relative relationships between joints and planes, the model is less likely to overfit to specific training examples and can better adapt to unseen data.
>
> $\textbf{Q3 What does ‘GAP’ mean?}$ We apologize for any confusion caused by the presence of two 'GAP' labels in $\textbf{Figure $\color{red}{2}$}$. You are correct in your assumption that 'GAP' stands for Global Average Pooling. We understand that this detail was not explicitly mentioned in the main manuscript, and we appreciate your attention to this matter. We have ensured that this is clarified in our revised manuscript.

---

> > ### Author Response · Authors · 2023-11-22
> > **Reply to Reviewer VfGb**
> >
> > $\textbf{Q4 How to generate the body/part attention map?}$ To generate the body/part attention map labels, we employ the inverse weak-perspective projection as described in the work by $\textit{Kissos et al. (2020)}$ [1]. The weak-perspective projection is based on the premise that the focal length and object distance are sufficiently large, allowing for the neglect of variations in the object along the Z-axis. It is a technique for converting three-dimensional camera coordinates into pixel coordinates.
> > We understand the importance of providing a clear explanation of this process, and we have included a more detailed introduction regarding the generation of body/part attention maps using inverse weak-perspective projection in the revised supplementary document (see $\textbf{Appendix.B. Part/Body Segmentation Maps}$ for more detail).
> > Since we generate 3D meshes using the ground truth (GT) SMPL labels and project them onto the pixel coordinate system, all body parts are visible. Even if a certain part is occluded in the image, we still use visible segmentation maps for supervision. This approach helps our attention mechanism effectively tackle the challenges posed by occlusions.
> >
> > $\textbf{References}$
> >
> > [1] Kissos I, Fritz L, Goldman M, et al. Beyond weak perspective for monocular 3d human pose estimation[C]//Computer Vision–ECCV 2020 Workshops: Glasgow, UK, August 23–28, 2020, Proceedings, Part II 16. Springer International Publishing, 2020: 541-554.

---

### Official Review · Reviewer_h8nb · 2023-10-22

**Soundness:** 3 good
**Presentation:** 1 poor
**Contribution:** 3 good
**Rating:** 5
**Confidence:** 5

**Summary:**

In this paper, the authors propose a reference-based reasoning network, known as ReBaR, for human pose and shape estimation, with a particular focus on handling occluded scenarios. As a single image-based method, ReBaR utilizes reference features for part regression reasoning to address challenges like occlusions and depth ambiguity. The paper presents extensive experiments to validate the effectiveness of the proposed method, and qualitative comparisons with previous state-of-the-art methods highlight significant improvements in handling depth ambiguity and occluded scenarios.

**Strengths:**

1. The paper addresses a more challenging setting of depth ambiguity and occluded scenarios, which is a commendable and important research direction.

2. ReBaR consistently outperforms existing state-of-the-art methods on multiple datasets, including AGORA, Human3.6M, 3DPW, and 3DPW-OCC.

3. The authors provide extensive qualitative comparisons and analyses, which effectively support the superiority of the proposed method over previous state-of-the-art techniques.

**Weaknesses:**

1. This paper should be better organized and improve the writing:

    The readability can be greatly enhanced if the author can further polish this paper, making long sentences and some of transactions more smooth.

    There are many typos that need to be corrected. For example:
       Last paragraph of the introduction: “datasets Then” should be “datasets. Then”.
      Section 3.3:   “ambiguity.Next” should be “ambiguity. Next”
     , etc…

2. Figures and tables should be clearer and better aligned with the text. For example:

    In Fig 2, the loss components should be more distinguishable. As indicated in the left corner, the auxiliary loss is marked as orange. However, all the losses are colored in orange.

    The relationships between elements in the figure and the main text should be clarified.

    The attention-guided encoders (AGE) are not indicated in Fig 2.

    It's unclear which components in Fig 2 are associated with the body-aware regression module.

    Please provide an explanation for the GAP component in Fig 2.

     It is suggested that the authors provide an overall architectural figure and detailed figures closely related to each section. Otherwise, readers need to retrieve Fig 2 frequently when reading the text.

    Table 1 would be better positioned adjacent to Section 4: Comparison to the State-of-the-Art.
Other improvements, such as those mentioned above, should also be addressed to enhance the paper's organization and clarity.

3. While the proposed method achieves excellent performance, it appears to be relatively complex. It would be beneficial to provide insights into the training process. Is the proposed method trained in multiple stages, such as initially training for 2D/3D pose estimation and subsequently training for the final mesh? Additionally, there are multiple loss components involved; it would be helpful to clarify whether these losses are optimized jointly or separately during different training stages.

4. The proposed method benefits from supervision by ground truth body/part segmentation maps, which some other methods do not utilize. To better understand the impact of this additional annotation, it would be valuable to conduct an ablation study. This would help ensure a fair and comprehensive comparison with other methods.


5. Given the potential computational cost of the proposed method, reporting the total parameters and actual inference times for each frame would be valuable information for readers and practitioners.

**Questions:**

see weakness

---

> ### Author Response · Authors · 2023-11-22
> **Reply to Reviewer h8nb**
>
> We have taken your comments into consideration and made the following improvements:
> 1. Restructured the content in the method section for better organization and flow.
> 2. Revised the writing to improve clarity and conciseness.
> 3. Enhanced the quality and alignment of figures and tables.
> 4. Corrected any typos and grammatical errors.
> These improvements significantly enhance the overall quality and presentation of our paper, making it more accessible and informative for readers. We truly appreciate your time and effort devoted to reviewing and providing valuable, detailed feedback on our paper.
>
> $\textbf{Q1 Providing insights into the training process.}$ To provide insights into the training process, we would like to clarify that our method optimizes the multiple loss components jointly almost throughout the training. However, we employ a small training trick borrowed from PARE to improve the effectiveness of the training process. Specifically, we train the model with all losses for the first 100 epochs. After that, we remove the segmentation supervision loss and continue training the model. This initial training phase helps the model learn to recognize and emphasize relevant regions in the input images. Once the attention mechanism has been guided towards the body parts, we remove the segmentation supervision loss and continue training. During this phase, the attention mechanism adapts further to better recover the human mesh from the images.
>
> $\textbf{Q2 To better understand the impact of body/part segmentation maps, it would be valuable to conduct an ablation study.}$ We greatly appreciate your meticulous suggestion to conduct an ablation study to better understand the impact of using the additional annotation provided by ground truth body/part segmentation maps.
> Therefore, we have supplemented our paper with additional experiments (see $\textbf{Section $\color{red}{4}$}$. Ablation Experiment for more detail) to enhance its rigor and ensure a fair and comprehensive comparison with other methods.
> To build a comparing method without using segmentation maps supervision (no region attention learning mechanism), we removed all attention-based learning component, and extend global features extracted from HRNet to align with the number of SMPL parts. Subsequently, we concatenate the T-Pose template SMPL parameters as tokens to the global features and predict SMPL parameters through a self-dimensionality reduction Transformer module. We refer this method as HMR + TF (no att.).
> As shown in the following table, the approach, which does not rely on attention-based methods, is at a disadvantage in terms of performance compared to our proposed method. This finding highlights the importance of the attention mechanism and the additional supervision provided by the ground truth body/part segmentation maps in achieving improved performance.
> |  Method  | MPJPE  | PA-MPJPE  | MPVPE  |
> |  ----  | ----  | ----  | ----  |
> | HMR (no att.) [1] |  85.1 | 52.2 | 118.5 |
> | PARE (+ part att.) [2] |  74.5 | 46.5 | 88.6 |
> | HMR + TF (no att.) |  76.8 | 49.0 |89.9 |
> | PARE + TF (+ part att.) | 73.2 | 44.7 | 85.0 |
> | ReBaR (+ part and body att.)  |  69.1 | 41.8 | 81.9 |
>
> $\textbf{Q3 Reporting the total parameters and actual inference times for each frame.}$ We provided the requested comparison results as follow, and added this information in our revised supplementary document (see $\textbf{Appendix.B. Inference Time and Parameter Counts}$ for more detail).
> |  Method  | PARAMS  | Inference Time  | Training Time  | MPJPE  | MPJPE_z  |
> |  ----  | ----  | ----  | ----  | ----  | ----  |
> | PARE  |  125.5MB | 1.24s | 16h | 74.5 | 58.2 |
> | PARE + TF  | 155.8MB | 1.32s | 19h | 73.2 | 56.6 |
> | ReBaR  |  222MB | 1.47s | 28h | 69.1 | 46.6 |
> | CLIFF* [3] | 305MB | 1.52s | / | 69.0 | 51.8 |
> | ReBaR*  | 361.1MB | 1.81s | 48h | 67.2 | 45.5 |
>
> $\textbf{References}$
>
> [1] Joo H, Neverova N, Vedaldi A. Exemplar fine-tuning for 3d human model fitting towards in-the-wild 3d human pose estimation[C]//2021 International Conference on 3D Vision (3DV). IEEE, 2021: 42-52.
>
> [2] Kocabas M, Huang C H P, Hilliges O, et al. PARE: Part attention regressor for 3D human body estimation[C]//Proceedings of the IEEE/CVF International Conference on Computer Vision. 2021: 11127-11137.
>
> [3] Li Z, Liu J, Zhang Z, et al. Cliff: Carrying location information in full frames into human pose and shape estimation[C]//European Conference on Computer Vision. Cham: Springer Nature Switzerland, 2022: 590-606.

---

### Official Review · Reviewer_nWz1 · 2023-10-30

**Soundness:** 3 good
**Presentation:** 2 fair
**Contribution:** 3 good
**Rating:** 6
**Confidence:** 4

**Summary:**

This paper proposes a novel method, named ReBaR, for human pose and shape estimation from single-view images. It includes two main modules: an attention-guided encoder and a body-aware regression module. The first module extracts features from both body and part regions using an attention-guided mechanism. Then, the second module uses a two-layer transformer to encode body-aware part features for each part and a one-layer transformer for per-part regression. Experimental results on two benchmark datasets show the effectiveness of the proposed method.

**Strengths:**

-The proposed method considers attention-based feature learning from both body and part regions which is reasonable for occlusion handling.

-The proposed method regresses additional relative depth to the torso plane, which helps reduce depth ambiguity.

-The experimental results show better performance than baseline methods.

**Weaknesses:**

1. The proposed method significantly outperforms the baseline method PARE in Table 1 and only slightly outperforms PARE in Table 2. Why does this happen?

2. To reduce depth ambiguity, the 3D GT labels such as SMPL parameters and 3D joints also provide the depth information of the human body. If they are accurately regressed, the depth information can be well estimated. Why is the relative depth loss L_{RD} still needed?

3. Some training details are unclear. For example, 1) what are the weight parameters for each loss? Please specify each weight parameter in the training; 2) considering there are so many losses, how to balance these losses in training? 3) how to generate the body/part attention map (M_p, M_B); 4) since some training sets may not have GT SMPL labels, how to ensure supervision? 5) The loss functions in Equation 3 are somewhat confusing. Please consider the consistency throughout the paper (e.g., L_{BS} and L_{PS} in Figure 2 appear to have the same meaning as L_{b_seg} and L_{p_seg} in Equation 3).

4. Considering transformers used in the architecture, how does the training/inference time of the proposed method compare to the baseline methods (PARE and CLIFF)?

5. What are the limitations and failure cases of the proposed method? A discussion is suggested to provide inspiration for future research directions.

6. Typos.
(1) On page 4, “R^{6840x3}” should be “R^{6890x3}”.
(2) On page 7, “Table 7 shows the result where our method still performs …” should be “Table 1”.
(3) On page 9, “MJE Depth reduced from 58.2 to 48.5” should be “… 58.2 to 47.5”

**Questions:**

Please refer to the weaknesses.

---

> ### Author Response · Authors · 2023-11-22
> **Reply to Reviewer nWz1**
>
> $\textbf{Q1 Why is the improvement difference between ReBaR and PARE in Table 1 and Table 2 so significant? }$ We appreciate your insightful comment and thank you for pointing out the need for clarification. The observed performance difference between the fine-tuned and non-fine-tuned comparisons can be attributed to the fine-tuning process. In our adjusted $\textbf{Table $\color{red}{1}$}$, we have separated the comparison into fine-tuned and non-fine-tuned results. The non-fine-tuned results of PARE on the AGORA dataset are consistent with those on the publicly available leaderboard, representing the performance of the model without fine-tuning. Consequently, ReBaR significantly outperforms PARE in the non-fine-tuned comparison.
> In response to your comment, we have also included the results of ReBaR without fine-tuning, allowing for a fair comparison with other models on the leaderboard that have not undergone fine-tuning. As demonstrated in the non-fine-tuned section of $\textbf{Table $\color{red}{1}$}$, among all methods employing weak-perspective projection, ReBaR still surpasses previous approaches even without fine-tuning.
> We also want to clarify that the performance improvement of ReBaR over PARE is significant. PARE achieved a result of PAMJE 46.5, while our method reached 41.8.
>
> $\textbf{Q2 Why is the relative depth loss $L_{RD}$ still needed? }$ Thank you for your insightful question. It is true that the 3D ground truth (GT) labels, such as SMPL parameters and 3D joints, provide depth information for the human body, and accurate regression can lead to a good estimation of depth. However, by defining three orthogonal planes and measuring the distance of a point to these planes, the relative depth loss captures the relative relationships between joints and the planes, which can further improve the performance due to the following possible reasons:
> 1. Local depth information: While regressing 3D GT labels primarily focuses on the global structure and pose of the human body, the relative depth loss emphasizes local depth information by considering the pairwise relationships between body joints and the planes. This helps the model to capture more subtle depth variations and improve the overall depth estimation.
> 2. Improved generalization: The relative depth loss  encourages the model to learn depth patterns that are more generalizable to different poses and body shapes. Since it focuses on the relative relationships between joints and planes, the model is less likely to overfit to specific training examples and can better adapt to unseen data.
>
> $\textbf{Q3 Some training details are unclear. }$ We apologize for any confusion regarding the training details. In this work, we follow the implementation of PARE for setting the loss weight parameters, where the weights for segmentation map supervision, reference feature 2D/3D auxiliary constraints, and relative depth supervision are set to 60, and the weights for SMPL parameter and keypoint supervision are five times greater (300). We have updated this information in the revised supplementary material (see $\textbf{Appendix.B. Loss Weight}$ for more detail).
>
> $\textbf{Q4 Since some training sets may not have GT SMPL labels, how to ensure supervision?}$ In order to ensure supervision when some training sets do not have ground truth (GT) SMPL labels, we follow the training practice of PARE. For the COCO dataset, we make use of pseudo-SMPL labels that are consistent with PARE to provide supervision. This allows us to train our model effectively even in the absence of genuine SMPL labels.
> For the 3DHP dataset, which SMPL labels is quietly coarse, we adapt our training strategy by not supervising the corresponding SMPL parameters and segmentation maps. Instead, we focus on supervising the keypoints, which are available in the dataset. This approach allows us to make the most of the available data and maintain supervision during the training process. For all other datasets in our training pipeline, we have access to genuine SMPL labels, which we use for supervision as intended.

---

> ### Author Response · Authors · 2023-11-22
> **Reply to Reviewer nWz1**
>
> $\textbf{Q5 How to generate the body/part attention map?}$ To generate the body/part attention map labels, we employ the inverse weak-perspective projection as described in the work by $\textit{Kissos et al. (2020)}$ [1]. The weak-perspective projection is based on the premise that the focal length and object distance are sufficiently large, allowing for the neglect of variations in the object along the Z-axis. It is a technique for converting three-dimensional camera coordinates into pixel coordinates.
> We understand the importance of providing a clear explanation of this process, and we have included a more detailed introduction regarding the generation of body/part attention maps using inverse weak-perspective projection in the revised supplementary document (see $\textbf{Appendix.B. Part/Body Segmentation Maps}$ for more detail).
>
> $\textbf{Q6 How does the training/inference time of the proposed method compare to the baseline methods (PARE and CLIFF)?}$ We appreciate your interest in comparing the training and inference time of our proposed method with the baseline methods, PARE and CLIFF. Here is a table summarizing the comparison results:
> |  Method  | PARAMS  | Inference Time  | Training Time  | MPJPE  | MPJPE_z  |
> |  ----  | ----  | ----  | ----  | ----  | ----  |
> | PARE[2]  |  125.5MB | 1.24s | 16h | 74.5 | 58.2 |
> | PARE + TF  | 155.8MB | 1.32s | 19h | 73.2 | 56.6 |
> | ReBaR[3]  |  222MB | 1.47s | 28h | 69.1 | 46.6 |
> | CLIFF*  | 305MB | 1.52s | / | 69.0 | 51.8 |
> | ReBaR*  | 361.1MB | 1.81s | 48h | 67.2 | 45.5 |
>
> Please note that since CLIFF does not provide open-source training code, we are unable to provide the training time for this method. We have included more detailed information about this comparison and the discussion in the revised supplementary document (see $\textbf{Appendix.B. Inference Time and Parameter Counts}$ for more detail).
>
> $\textbf{Q7 The limitations and failure cases.}$ We appreciate your suggestion to discuss the limitations and failure cases of our proposed method, ReBaR. We have included failure case examples and a detailed discussion of these aspects in the revised supplementary document (see$\textbf{Appendix.E}$  for more detail) and updated the conclusion section to mention limitations and future works as well.
>
> $\textbf{We have rectified all writing errors in the paper, standardized the notation, and prevented any confusion for the readers.
> }$
>
> $\textbf{References}$
>
> [1] Kissos I, Fritz L, Goldman M, et al. Beyond weak perspective for monocular 3d human pose estimation[C]//Computer Vision–ECCV 2020 Workshops: Glasgow, UK, August 23–28, 2020, Proceedings, Part II 16. Springer International Publishing, 2020: 541-554.
>
> [2] Kocabas M, Huang C H P, Hilliges O, et al. PARE: Part attention regressor for 3D human body estimation[C]//Proceedings of the IEEE/CVF International Conference on Computer Vision. 2021: 11127-11137.
>
> [3] Li Z, Liu J, Zhang Z, et al. Cliff: Carrying location information in full frames into human pose and shape estimation[C]//European Conference on Computer Vision. Cham: Springer Nature Switzerland, 2022: 590-606.

---

### Official Review · Reviewer_mMLH · 2023-10-31

**Soundness:** 2 fair
**Presentation:** 1 poor
**Contribution:** 2 fair
**Rating:** 5
**Confidence:** 5

**Summary:**

Authors introduced the novel method:  reference-based reasoning for robust human 3D mesh reconstruction task. Authors proposed to extract features from both body and part regions using attention-guided mechanism. Then they proposed the reference based reasoning by inferring the spatial relationships of occluded parts with the body. Experiments are conducted on two benchmarks and demonstrated the goo d performance.

**Strengths:**

Authors are tackling the challenging problem: 3D human mesh reconstruction in challenging cases when the severe occlusion exists in RGB images.
The idea of encoding part regions and global regions together looks interesting.

**Weaknesses:**

Less qualitative results for targetted scenario: Though authors insist that they tackle the problem of severe occlusion and depth ambiguity; while the results presented in the main result is not showing the results for that. Especially, the figure in the 5th row of the Fig. 4 is neigher occluded nor confused in 3d depth. Other results are also not including challenging cases. I think authors need to show the specific cases they are intending.
Also, the compared algorithms are not sufficient enough to validate the effectiveness of the method. The ProPose algorithm (CVPR’23) is shown in the Fig. 1; while it is not compared in the Figure 4. Also, in terms of PAMJE, the proposed method is not clearly the SOTA in Table 2.
Explanations are also not clear, especially, the usage of the losses in Sec. 3.4. In Sec. 3.2 authors explained their attention is learned using part-segmentation maps; while in Sec. 3.4, seemingly all the losses are applied to the entire network. I think authors need to clarify the aspect. Also, some `hat’ notations in Sec. 3.4 are wrongly used.

**Questions:**

Did authors made the part segmentation loss only applied to the attention learning? or other losses are also applied to learn it?

---

> ### Author Response · Authors · 2023-11-22
> **Reply to Reviewer mMLH**
>
> $\textbf{Q1 Less qualitative results for targetted scenario.}$ In response to your concern regarding the lack of qualitative results for targeted scenarios, we have made significant updates to our supplementary material and revised paper to address this issue. We have included many challenging cases from 3DPW-OCC, LSPET, and in-the-wild datasets, where our method, ReBaR, demonstrates superior performance compared to existing methods, particularly on the in-the-wild dataset.
> Following your suggestion, we have restructured the comparison and incorporated ProPose into our revised paper. All images presented are sourced from the LSPET dataset. As shown in the updated $\textbf{Figure $\color{red}{4}$}$ (in $\textbf{Section $\color{red}{4}$}$), our method outperforms existing approaches in complex scenarios and challenging poses that are commonly found in motion datasets. This is especially true in cases involving occlusion and depth ambiguity.
> We believe that this thorough evaluation effectively highlights the advantages of our approach in handling intricate and challenging cases. We hope that these updates adequately address your concerns and demonstrate the effectiveness of ReBaR in tackling the problem of severe occlusion and depth ambiguity.
>
> $\textbf{Q2 In terms of PAMJE, the proposed method is not clearly the SOTA in Table 2.}$ In response to your comment regarding the PAMJE metric in $\textbf{Table $\color{red}{2}$}$, we would like to clarify that our method does achieve state-of-the-art performance. While the difference between the current best result of 40.6 and our method's result of 40.8 may appear small.
> Furthermore, our method outperforms all existing methods in terms of other key metrics, such as MJE and MVE. This demonstrates the overall effectiveness and superiority of our approach compared to existing state-of-the-art methods.
> We believe that these results, combined with our method's robust performance in challenging scenarios involving occlusion and depth ambiguity, as mentioned in our previous response, showcase the value and contribution of our work in this field.
>
> $\textbf{Q3 The explanation of the relationship between objective function updating and attention learning is unclear.}$ We apologize for any confusion that may have arisen from our explanations in $\textbf{Sections $\color{red}{3.2}$}$ and $\textbf{Sections $\color{red}{3.4}$}$ of the original manuscript. We appreciate your feedback and have made necessary clarifications in the revised manuscript.
> In $\textbf{Sections $\color{red}{3.2}$}$ of the original manuscript, our intention was to emphasize that the attention mechanism is auxiliary supervised by part-segmentation maps. However, we understand that our previous description might have led to the impression that it is solely supervised by the segmentation maps. We have corrected this description in the revised manuscript (see $\textbf{Sections $\color{red}{3.1}$}$ for more detail) to avoid any further misunderstanding.
> However, it is important to note that the part-segmentation loss is not the only loss employed in our method. In $\textbf{Sections $\color{red}{3.3}$}$ of the revised paper, the losses are indeed applied to the entire network, but the attention mechanism's supervision with part-segmentation maps is an additional component that aids in the learning process. This auxiliary supervision helps improve the performance of our method in handling challenging cases, such as occlusions and depth ambiguities.
> We hope that these clarifications address your concerns and provide a better understanding of our method and its implementation.
>
>
> $\textbf{All writing errors have been corrected in the revised version of the paper.}$

---

### Official Review · Reviewer_aFNK · 2023-10-31

**Soundness:** 3 good
**Presentation:** 3 good
**Contribution:** 2 fair
**Rating:** 5
**Confidence:** 5

**Summary:**

To address the problem of occlusion and depth ambiguity, this paper proposes ReBaR, which learns reference features for part regression reasoning. This results show that it can outperform baseline methods, especially in the evaluation of depth.

**Strengths:**

1. The motivation is clear and interesting.
2. Experiments are extensive and achieve state-of-the-art results.

**Weaknesses:**

1. The approach section contains too many details and lacks motivations and general pictures. Also, the pipeline is blurry.
2. Is PARE your baseline? If so, I am interested in whether REBAR and PARE have the same experimental setting and the comparison in FLOPS and PARAMS, especially for the PARAMS in the part branch.
3. It seems not very clear to me why your method achieves better results in depth with your network design.

The core question is why does your method perform better, especially in depth compared with PARE? Is that because the increase of data or parameters or the network design.

**Questions:**

Please see weakneeses

---

> ### Author Response · Authors · 2023-11-22
> **Reply to Reviewer aFNK**
>
> $\textbf{Q1 Improve the approach section and supplement the motivation and general picture.}$  Thank you for taking the time to review our work and provide valuable feedback. We appreciate your concerns regarding the approach section.
> We understand that the approach section may seem dense with details, but we believe that providing a comprehensive description of the methods is essential for the reproducibility of our results. It allows fellow researchers to better understand the nuances of our work and build upon it in the future. However, we acknowledge that motivations and general pictures are crucial to provide context and make our work more accessible.
> In light of your feedback, we revised the approach section (see $\textbf{Section $\color{red}{3}$}$ for more detail) to include more motivations and made significant improvements to the pipeline (see $\textbf{Figure $\color{red}{2}$}$) to present the framework more clearly. We believe that these modifications will help readers grasp the overall concept and purpose of our method design.
> Once again, thank you for your constructive feedback. We hope that our $\textbf{revisions }$will address your concerns while maintaining the integrity of our methods and results.
>
> $\textbf{Q2 The FLOPS and PARAMS of ReBaR and PARE (especially in part branch).}$ Yes, PARE serves as our baseline, and both REBAR and PARE share the same experimental setting. The part branch in REBAR has identical PARAMS and FLOPS as those in PARE. To provide a clear comparison, we present the FLOPS, PARAMS, and Inference Time for each method in the table below:
> |  Method  | FLOPS  | PARAMS  | Inference Time  | MPJPE  | MPJPE_z  |
> |  ----  | ----  | ----  | ----  | ----  | ----  |
> | PARE[1]  | 14.9G | 125.5MB | 1.24s | 74.5  | 58.2  |
> | PARE + TF  | 18.1G | 155.8MB | 1.32s | 73.2  | 56.6  |
> | ReBaR  | 26.4G | 222MB | 1.47s | 69.1  | 46.6  |
>
> Our approach enhances performance while maintaining an acceptable inference speed.
>
> $\textbf{Q3 Why does your method perform better, especially in depth compared with PARE?}$ We appreciate the reviewer's concern about the improved performance in our method compared to PARE. The primary reason for our method's better performance, especially in depth estimation, is the introduction of body-aware features and the use of a transformer to model their dependencies.
> Our method incorporates an attention-guided encoder (AGE) and a body-aware regression module (BAR), which are specifically designed to capture both global and local features of the human body and establish relationships between different body parts. The key difference between our method and PARE is the learning of body-aware features and the use of a transformer to model their dependencies.
> To better illustrate the impact of body-aware features on the depth estimation, we provided a comparison as follow between PARE, PARE+TF (a modified version of our method without body-aware features), and our method. The results show that the main reason for the performance improvement is the introduction of body-aware features into the transformer learning.
> |  Method  | MPJPE  | PAMPJPE  | MPVPE  | MPJPE_z  |
> |  ----  | ----  | ----  | ----  | ----  |
> | PARE  | 74.5 | 46.5 | 88.6 |58.2  |
> | PARE + TF  | 73.2 | 44.7 | 85.0 | 56.6  |
> | PARE + Ba-Feat + TF  | 70.4 | 42.4 | 82.2 | 47.5  |
>
> By incorporating body-aware features, our method can better understand the spatial relationships between body parts and the whole body, leading to a more accurate 3D reconstruction, especially in depth. The use of a transformer allows our model to effectively exploit the spatial relationships between parts and bodies, generating more informative part representations and improving depth estimation.
> In summary, the improved depth estimation in our method is primarily attributed to the incorporation of body-aware features and the use of a transformer to model their dependencies. This key difference in our method's design principles plays a significant role in achieving better depth estimation compared to PARE.
>
> $\textbf{References}$
>
> [1] Kocabas M, Huang C H P, Hilliges O, et al. PARE: Part attention regressor for 3D human body estimation[C]//Proceedings of the IEEE/CVF International Conference on Computer Vision. 2021: 11127-11137.

---

### Public Comment · ~Jinhe_huang1 · 2023-12-10
**To dear Authors**

Dear authors,

Based on the provided results and video, it appears that REBAR is the first method I've encountered that effectively addresses the issue of arm depth ambiguity. This is a significant advancement in motion capture and could be of great help to me. I would like to inquire if you have any plans to release the source code in the future? Looking forward to your response.

---

### Meta-Review · Area_Chair_1Chj · 2023-12-05

**Metareview:**

The authors propose a method for human shape and pose estimation from a single image, based on encoding images into body features and part features. These features are then regressed to camera parameters and SMPL body parameters using a transformer. The authors evaluate on commonly used 3D human datasets, such as 3DPW and AGORA (with fine-tuning). The model is trained on Human3.6M, COCO-EFT, MPII, and MPI-INF-3DHP. While the qualitative results shown by the authors are promising, the relative quantitative improvement is small, in addition to the method being more computationally expensive compared to baselines. As a result, it is difficult to evaluate if the model improvements are due to a larger parameter space or the technical contributions of the method. All reviewers have raised similar concerns about the results. The authors are encouraged to revise their paper based on reviewer feedback, especially regarding concerns not addressed in the rebuttal (ex: relationship between losses on the 3D joints loss and relative depth, experiments regarding model size). I would also encourage the authors to perform evaluations on other downstream tasks or datasets with potentially more occlusion (ex: curate a subset of 3DPW, or show the error distribution of occluded vs. non-occluded parts) to better understand the method performance.

**Justification For Why Not Higher Score:**

I appreciate the response from the authors and discussion with reviewers. However, this method introduces a lot of complexity in training and model design, and the performance improvement is relatively small given the additional computation/time required during training and inference. It is also unclear if the model simply performs better due to having more parameters than baseline. Additionally, there are a large set of losses considered by the authors during training in Section 3.3 and there is no ablation on the loss terms or how these terms might be weighted/interfere with each other during training. Most reviewers have remaining concerns after the rebuttal as well.

**Justification For Why Not Lower Score:**

N/A

---

### Decision · Program_Chairs · 2024-01-16

Reject